# Diffusion Tuning: Transferring Diffusion Models via Chain of Forgetting

**Jincheng Zhong,*  Xingzhuo Guo,*  Jiaxiang Dong, Mingsheng Long**$^{\boxtimes}$

School of Software, BNRist, Tsinghua University, China

{zjc22,gxz23,djx20}@mails.tsinghua.edu.cn, mingsheng@tsinghua.edu.cn

## Abstract

Diffusion models have significantly advanced the field of generative modeling. However, training a diffusion model is computationally expensive, creating a pressing need to adapt off-the-shelf diffusion models for downstream generation tasks. Current fine-tuning methods focus on parameter-efficient transfer learning but overlook the fundamental transfer characteristics of diffusion models. In this paper, we investigate the transferability of diffusion models and observe a monotonous *chain of forgetting* trend of transferability along the reverse process. Based on this observation and novel theoretical insights, we present *Diff-Tuning*, a frustratingly simple transfer approach that leverages the chain of forgetting tendency. Diff-Tuning encourages the fine-tuned model to retain the pre-trained knowledge at the end of the denoising chain close to the generated data while discarding the other noise side. We conduct comprehensive experiments to evaluate Diff-Tuning, including the transfer of pre-trained Diffusion Transformer models to eight downstream generations and the adaptation of Stable Diffusion to five control conditions with ControlNet. Diff-Tuning achieves a 24.6% improvement over standard fine-tuning and enhances the convergence speed of ControlNet by 24%. Notably, parameter-efficient transfer learning techniques for diffusion models can also benefit from Diff-Tuning. Code is available at this repository: https://github.com/thuml/Diffusion-Tuning.

## 1  Introduction

Diffusion models [45, 17, 47] are leading the revolution in modern generative modeling, achieving remarkable successes across various domains such as image [11, 39, 12], video [43, 19, 55], 3D shape [34], audio generation [25], etc. Despite these advances, training an applicable diffusion model from scratch often demands a substantial computational budget, exemplified by the thousands of TPUs needed, as reported by [55]. Consequently, fine-tuning well pre-trained, large-scale models for specific tasks has become increasingly crucial in practice [54, 60, 56].

During the past years, the deep learning community has concentrated on how to transfer knowledge from large-scale pre-trained models with minimal computational and memory demands, a process known as parameter-efficient fine-tuning (PEFT) [20, 59, 54, 7, 21, 32]. The central insight of these approaches is to update as few parameters as possible while avoiding performance decline. However, the intrinsic transfer properties of diffusion models have remained largely unexplored, with scant attention paid to effectively fine-tuning from a pre-trained diffusion model.

Previous studies on neural network transferability, such as those by [33, 57], have demonstrated that lower-level features are generally more transferable than higher-level features. In the context of diffusion models, which transform noise into data through a reverse process, it is logical to assume

---

*Equal contribution

38th Conference on Neural Information Processing Systems (NeurIPS 2024).

that the initial stages, which are responsible for shaping high-level objects, differ in transferability from later stages that refine details. This differential transferability across the denoising stages presents an opportunity to enhance the efficacy of fine-tuning.

In this work, we investigate the transferability within the reverse process of diffusion models. Firstly, we propose that a pre-trained model can act as a universal denoiser for lightly corrupted data, capable of recognizing and refining subtle distortions (see Figure 1). This ability leads to improved generation quality when we directly replace the fine-tuned model with the original pre-trained one under low distortion. The suboptimality observed with fine-tuned models suggests potential overfitting, mode collapse, or undesirable forgetting. Then we extend the experiments by gradually increasing the denoising steps replaced, to cover higher-level noised data, observing the boundaries of zero-shot generalization capability. This indicates that the fine-tuning objective should prioritize high-level shaping, associated with domain-specific characteristics. We term this gradual loss of adaptability the *chain of forgetting*, which tends to retain low-level denoising skills while forgetting high-level, domain-specific characteristics during the transfer of the pre-trained model. We further provide novel theoretical insights to reveal the principles behind the chain of forgetting.

Since the chain of forgetting suggests different denoising stages lead to different forgetting preferences, it is reasonable to develop a transfer strategy that balances the degrees of forgetting and retention. Technically, based on the above motivation, we propose *Diff-Tuning*, a frustratingly simple but general fine-tuning approach for diffusion models. Diff-Tuning extends the conventional fine-tuning objectives by integrating two specific aims: 1) knowledge retention, which retains general denoising knowledge; 2) knowledge reconsolidation, which tailors high-level shaping characteristics to specific downstream domains. Diff-Tuning leverages the chain of forgetting to balance these two complementary objectives throughout the reverse process.

Experimentally, Diff-Tuning achieves significant performance improvements over standard fine-tuning in two mainstream fine-tuning scenarios: conditional generation and controllable generation with ControlNet [60]. Our contributions can be summarized as follows:

- Motivated by the transferable features of deep neural networks, we explore the transferability of diffusion models through the reverse process and observe a chain of forgetting tendency. We provide a novel theoretical perspective to elucidate the underlying principles of this phenomenon for diffusion models.

- We introduce Diff-Tuning, a frustratingly simple yet effective transfer learning method that integrates two key objectives: knowledge retention and knowledge reconsolidation. Diff-Tuning harmonizes these two complementary goals by leveraging the chain of forgetting.

- As a general transfer approach, Diff-Tuning achieves significant improvements over its standard fine-tuning counterparts in conditional generation across eight datasets and controllable generation using ControlNet under five distinct conditions. Notably, Diff-Tuning enhances the transferability of the current PEFT approaches, demonstrating the generality.

## 2 Related Work

### 2.1 Diffusion Models

Diffusion models [17] and their variants [47, 48, 23] represent the state-of-the-art in generative modeling [12, 3], capable of progressively generating samples from random noise through a chain of denoising processes. Researchers have developed large-scale foundation diffusion models across a broad range of domains, including image synthesis [17], video generation [19], and cross-modal generation [43, 42]. Typically, training diffusion models involves learning a parametrized function $f$ to distinguish the noise signal from a disturbed sample, as formalized below:

$$L(\theta) = \mathbb{E}_{t, \mathbf{x}_0, \boldsymbol{\epsilon}} \left[ \left\| \boldsymbol{\epsilon} - f_\theta \left( \sqrt{\alpha_t} \mathbf{x}_0 + \sqrt{1 - \alpha_t} \boldsymbol{\epsilon}, t \right) \right\|^2 \right] \tag{1}$$

where $\mathbf{x}_0 \sim \mathcal{X}$ represents real samples, $\boldsymbol{\epsilon} \sim \mathcal{N}(\mathbf{0}, \mathbf{I})$ denotes the noise signal, and $\mathbf{x}_t = \sqrt{\alpha_t} \mathbf{x}_0 + \sqrt{1 - \alpha_t} \boldsymbol{\epsilon}$ is the disturbed sample at timestep $t$. Sampling from diffusion models follows a Markov chain by iteratively denoising from $\mathbf{x}_T \sim \mathcal{N}(\mathbf{0}, \mathbf{I})$ to $\mathbf{x}_0$.

Previous research on diffusion models primarily focuses on noise schedules [36, 23], training objectives [44, 23], efficient sampling [46], and model architectures [39]. In contrast to these existing

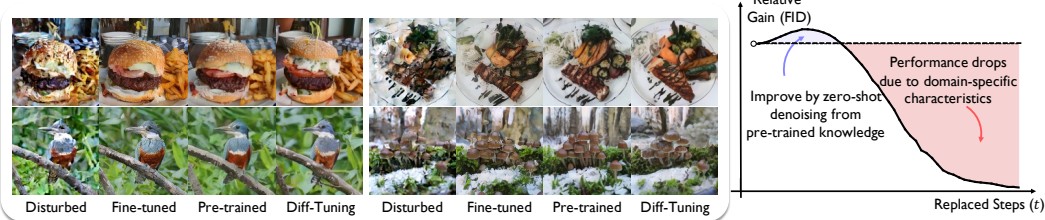

Figure 1: Case study of directly replacing the denoiser with the original pre-trained model on lightly disturbed data (left). The changes in Fréchet Inception Distance (FID) as the denoising steps are incrementally replaced by the original pre-trained model (right).

works, our method investigates the transferability of diffusion models across different denoising stages and enhances the transfer efficacy in a novel and intrinsic way.

## 2.2 Transfer Learning

Transfer learning [38] is an important machine learning paradigm that aims to improve the performance of target tasks by leveraging knowledge from source domains. Transferring from pre-trained models, commonly known as fine-tuning, has been widely proved effective in practice, especially for the advanced large-scale models [5, 1, 12]. However, directly fine-tuning a pre-trained model can cause overfitting, mode collapse, and catastrophic forgetting [24]. Extensive prior work has focused on overcoming these challenges to ultimately enhance the utilization of knowledge from pre-trained models [2, 8, 61]. However, effective transfer of diffusion models has received scant attention.

**Parameter-Efficient Fine-tuning (PEFT)**   With significant advancements in the development of large-scale models [10, 5, 1, 12], research in transfer learning has increasingly concentrated on PEFT methods that minimize the number of learnable parameters. The primary goal of PEFT is to reduce time and memory costs associated with adapting large-scale pre-trained models. Techniques such as incorporating extra adapters [20, 60, 35] and learning partial or re-parameterized parameters [59, 21, 22, 14] are employed for their effectiveness in reducing computational demands. Nevertheless, the reliance on deep model architectures and the necessity of carefully selecting optimal placements present substantial challenges. Intuitively, PEFT approaches could potentially mitigate catastrophic forgetting by preserving most parameters unchanged; for a detailed discussion, refer to Section 4.3.

**Mitigating Catastrophic Forgetting**   Catastrophic forgetting is a long-standing challenge in the context of continual learning, lifelong learning, and transfer learning, referring to the tendency of neural networks to forget previously acquired knowledge when fine-tuning on new tasks. Recent exploration in parameter regularization approaches [24, 28, 27, 8] have gained prominence. Approaches such as [58, 29, 52] propose the data-based regularization, which involves distilling pre-trained knowledge into a knowledge bank. However, efforts to mitigate forgetting within the framework of diffusion models remain notably scarce.

## 3 Method

### 3.1 Chain of Forgetting

Compared with one-way models, diffusion models specify in a manner of multi-step denoising and step-independent training objectives. Inspired by prior studies on the transferability of deep neural features [57, 33], we first explore how the transferability of diffusion models varies along the denoising steps.

**Pre-trained Model Serves as a Zero-Shot Denoiser**   Modern large-scale models are pre-trained with a large training corpus, emerging powerful zero-shot generalization capabilities. We begin by analyzing whether the pre-trained diffusion models hold similar zero-shot denoising capabilities. In particular, we utilize a popular pre-trained Diffusion Transformer (DiT) model [39] as our testbed. We fine-tune the DiT model on a downstream dataset. When the reverse process comes to the the last 10%

steps, we switch and continue the remaining denoising steps with the fine-tuned model, the original pre-trained model, and our Diff-Tuning model respectively. We visualize a case study in Figure 1 (left) with corresponding replacement setups. Surprisingly, the results reveal that replacement by the pre-trained model achieves competitive quality, even slightly better than the fine-tuned one, indicating that the pre-trained diffusion model indeed holds the zero-shot denoising skills. On the other side, some undesirable overfitting and forgetting occur when fine-tuning diffusion models.

**Forgetting Trend**   Next, we delve deeper into investigating the boundary of generalization capabilities for the pre-trained model. Figure 1 (right) illustrates the performance trend when we gradually increase the percentage of denoising steps replaced from $0$ to $100\%$. Initially, this naive replacement yields better generation when applied towards the end of the reverse process. However, as more steps are replaced, performance begins to decline due to domain mismatch. This trend suggests the fine-tuned model may overfit the downstream task and forget some of the fundamental denoising knowledge initially possessed by the pre-trained model when $t$ is small. Conversely, as $t$ increases, the objects desirable in the new domain are distorted by the pre-trained model, resulting in a performance drop. Based on these observations, we conceptually separate the reverse process into two stages: (1) domain-specific shaping, and (2) general noise refining. We claim that the general noise refining stage is more transferable and can be reused across various domains. In contrast, the domain-specific shaping stage requires the fine-tuned model to forget the characteristics of the original domain and relearn from the new domains.

**Theoretic Insights**   Beyond empirical observations, we provide a novel theoretical perspective of the transfer preference for the pre-trained diffusion model. Following the objectives of diffusion models, a denoiser $F$ (an $\mathbf{x}_0$-reparameterization [23] of $f$ in Eq. (1)) is to approximate the posterior expectation of real data over distribution $\mathcal{D}$. This is formalized by:

$$F(\mathbf{x}_t) = \mathbb{E}_{\mathbf{x}_0 \sim p(\mathbf{x}_0|\mathbf{x}_t)} \left[ \mathbf{x}_0 \right] = \frac{\int_{\mathbf{x}_0} \mathcal{N}\left(\mathbf{x}_t; \sqrt{\alpha_t}\mathbf{x}_0, (1-\alpha_t)\mathbf{I}\right) \cdot \mathbf{x}_0 \cdot p_{\mathcal{D}}(\mathbf{x}_0)\mathrm{d}\mathbf{x}_0}{\int_{\mathbf{x}_0} \mathcal{N}\left(\mathbf{x}_t; \sqrt{\alpha_t}\mathbf{x}_0, (1-\alpha_t)\mathbf{I}\right) \cdot p_{\mathcal{D}}(\mathbf{x}_0)\mathrm{d}\mathbf{x}_0}, \tag{2}$$

where $p_{\mathcal{D}}(\mathbf{x}_0)$ represents the distribution of real data from $\mathcal{D}$, and $\mathcal{N}$ denotes the Gaussian distributions determined by the forward process. Notably, a larger variance of Gaussian distribution indicates a more uniform distribution. Through a detailed investigation of these Gaussian distributions under varying timesteps $t$, we derive the following theorem. All proofs and derivations are provided in Appendix A.

**Theorem 1 (Chain of Forgetting)** *Suppose a diffusion model with $\lim\limits_{t \to 0} \alpha_t = 1$ and $\lim\limits_{t \to T} \alpha_t = 0$ over finite samples, then the ideal denoiser $F$ satisfies*

  1. $\lim\limits_{t \to 0} F(\mathbf{x}_t) = \operatorname*{argmin}\limits_{p(\mathbf{x}_0)>0}\{\|\mathbf{x}_0 - \mathbf{x}_t\|\}$, *i.e., the closest sample in dataset.*

  2. $\lim\limits_{t \to T} F(\mathbf{x}_t) = \mathbb{E}_{\mathbf{x}_0 \sim p_{\mathcal{D}}(\mathbf{x}_0)}[\mathbf{x}_0]$, *i.e., the mean of data distribution.*

Theorem 1 elucidates the mechanism behind the chain of forgetting. On one hand, when $t \to 0$, a model optimized on a training dataset $\mathcal{D}$ can perform zero-shot denoising within the vicinity of the support set supp$(\mathcal{D})$. As the training dataset scale expands, so does the coverage of supp$(\mathcal{D})$, enabling diffusion models to act as general zero-shot denoisers for data associated with small $t$. On the other hand, as $t \to T$, the model's generalization is significantly influenced by the distribution distance dist$(\mathbb{E}_{\mathcal{D}}[\mathbf{x}_0], \mathbb{E}_{\mathcal{D}_{\text{new}}}[\mathbf{x}_0^{\text{new}}])$, where $\mathcal{D}_{\text{new}}$ denotes the dataset of the new domain. This theorem highlights the necessity for further adaptation in the new domain.

## 3.2   Diff-Tuning

Based on the above observations and theoretical insights, we introduce Diff-Tuning, which incorporates two complementary strategies to leverage the chain of forgetting in the reverse process: 1) knowledge retention, and 2) knowledge reconsolidation. Diff-Tuning aims to retain general denoising skills from the pre-trained model while discarding its redundant, domain-specific shaping knowledge. This enables the model to adapt more effectively to the specific characteristics of downstream tasks. Diff-Tuning harmonizes the retention and reconsolidation via the chain of forgetting tendency.

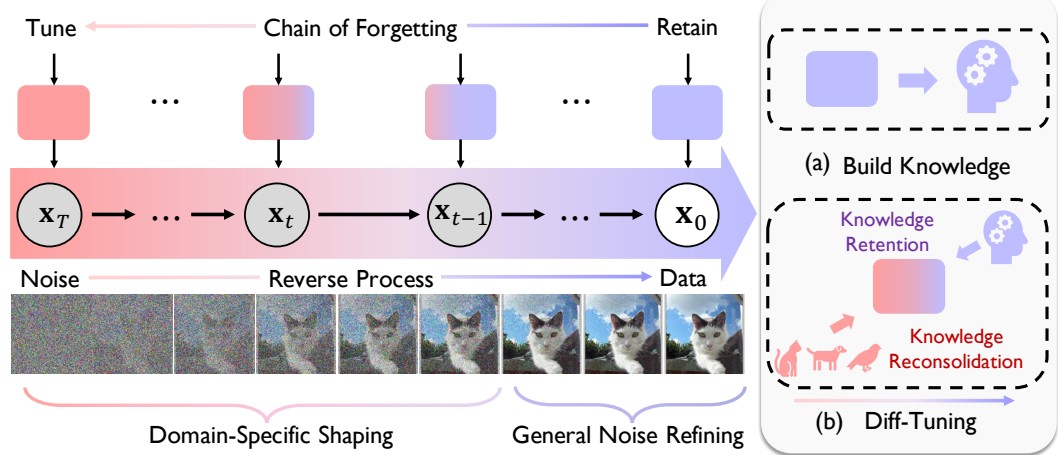

Figure 2: The conceptual illustration of the chain of forgetting (Left). The increasing forgetting tendency as $t$ grows. (a) Build a knowledge bank for the pre-trained model before fine-tuning. (b) Diff-Tuning leverages knowledge retention and reconsolidation, via the chain of forgetting.

Without loss of generality, we present Diff-Tuning under the standard DDPM objective, omitting conditions in the formulations. The general conditional generation setup will be discussed later.

**Knowledge Retention** As discussed earlier, retaining pre-trained knowledge during the latter general noising refining proves beneficial. However, the classic parameter-regularization-based approaches [24, 8, 27] mitigate forgetting uniformly across the reverse process, primarily due to the parameter-sharing design inherent in diffusion models. To address this, Diff-Tuning constructs an augmented dataset $\widehat{\mathcal{X}}^s = \{\widehat{\mathbf{x}}^s, \cdots\}$, pre-sampled from the pre-trained model. This dataset acts as a repository of the retained knowledge of the pre-trained model. We define the auxiliary training objective, $L_{\text{retention}}$, as follows:

$$L_{\text{retention}}(\theta) = \mathbb{E}_{t,\boldsymbol{\epsilon},\widehat{\mathbf{x}}_0^s \sim \widehat{\mathcal{X}}^s} \left[ \xi(t) \left\| \boldsymbol{\epsilon} - f_\theta \left( \sqrt{\alpha_t}\widehat{\mathbf{x}}_0^s + \sqrt{1-\alpha_t}\boldsymbol{\epsilon}, t \right) \right\|^2 \right], \tag{3}$$

where $\xi(t)$ is the retention coefficient. In accordance with the principles of the chain of forgetting, $\xi(t)$ decreases monotonically with increasing $t$, promoting the retention of knowledge associated with small $t$ values and the discarding of knowledge related to large $t$ values. Knowledge Retention shares a similar formulation with the pre-training objective but without the reliance on the original pre-training dataset.

It is important to note that the concept of Knowledge Retention in Diff-Tuning is anchored in the principle of the chain of forgetting. This approach encourages the model to recall how to denoise samples with low levels of disturbance, as reflected by the retention coefficient $\xi(t)$. While the proposed augmented dataset serves as an easy-to-implement example of this concept, given the flexibility of Diff-Tuning, various other methods can also effectively facilitate knowledge retention. An alternative approach involving knowledge distillation is detailed in Appendix C.

**Knowledge Reconsolidation** In contrast to knowledge retention, knowledge reconsolidation focuses on adapting pre-trained knowledge to new domains. The intuition behind knowledge reconsolidation is to diminish the conflict between forgetting and adaptation by emphasizing the tuning of knowledge associated with large $t$. This adaptation is formalized as follows:

$$L_{\text{adaptaion}}(\theta) = \mathbb{E}_{t,\boldsymbol{\epsilon},\mathbf{x}_0 \sim \mathcal{X}} \left[ \psi(t) \left\| \boldsymbol{\epsilon} - f_\theta \left( \sqrt{\alpha_t}\mathbf{x}_0 + \sqrt{1-\alpha_t}\boldsymbol{\epsilon}, t \right) \right\|^2 \right], \tag{4}$$

where $\psi(t)$ is the reconsolidation coefficient, a monotonic increasing function within the range $[0, 1]$, reflecting increased emphasis on domain-specific adaptation as $t$ increases. Similar to Knowledge Retention, the principle behind Knowledge Reconsolidation involves adapting the model to effectively

handle samples that are heavily disturbed, which are more significantly influenced by the distance between the pre-trained and target distributions. Reweighting by $\psi(t)$ serves as one of the simplest implementations of this concept.

**A Frustratingly Simple Approach**  Overall, we reach Diff-Tuning, a general and flexible fine-tuning framework for effective transferring pre-trained diffusion models to downstream generations, the overall objective is as follows:

$$\min_{\theta} L_{\text{retention}}(\theta) + L_{\text{adaptation}}(\theta), \tag{5}$$

where $L_{\text{retention}}(\theta)$ and $L_{\text{adaptation}}(\theta)$ are described before, $\theta$ represents the set of tunable parameters. Notably, Diff-Tuning is architecture-agnostic and seamlessly integrates with existing PEFT methods. Further details are discussed in Section 4.3.

**Choices of $\xi(t)$ and $\psi(t)$**  For clarity and simplicity, we define $\xi(t) = 1 - \psi(t)$, ensuring equal weighting for each $t$, following the original DDPM configuration. This complementary design excludes the influences of recent studies on the $t$-reweighting techniques [23, 12, 9]. From the above discussion, we can choose any monotonically increasing function whose range falls in $[0, 1]$. In this work, we scale the variable $t$ to the interval $[0, 1]$, and apply a simple power function group $\psi(t) = t^{\tau}$ for practical implementation. In our experiments, we report the main results with $\tau = 1$, and the variations of the choice are explored in Section 4.4.

**Conditional Generation**  Classifier-free guidance (CFG) [18] forms the basis for large-scale conditional diffusion models. To facilitate sampling with CFG, advanced diffusion models such as DiT [39] and Stable Diffusion [12] are primarily trained conditionally. CFG is formulated as $\boldsymbol{\epsilon} = (1 + w)\boldsymbol{\epsilon}_{\text{c}} - w\boldsymbol{\epsilon}_{u}$, where $w, \boldsymbol{\epsilon}_c, \boldsymbol{\epsilon}_u$ are the CFG weight, conditional output, and unconditional output. As a general approach, Diff-Tuning inherits the conditional training and sampling setup to support a wide range of transfer tasks. Due to the mismatch between the pre-training domain and downstream tasks in the conditional space, we apply knowledge retention $L_{\text{retention}}$ on the unconditional branch and knowledge reconsolidation $L_{\text{adaptation}}$ on both unconditional and conditional branches.

## 4 Experiments

To fully verify the effectiveness of Diff-Tuning, we extensively conduct experiments across two mainstream fine-tuning scenarios: 1) Class-conditional generation, which involves eight well-established fine-grained downstream datasets, and 2) Controllable generation using the recently popular ControlNet [60], which includes five distinct control conditions.

### 4.1 Transfer to Class-conditional Generation

**Setups**  Class-conditioned generation is a fundamental application of diffusion models. To fully evaluate transfer efficiency, we adhere to the benchmarks with a resolution of $256 \times 256$ as used in DiffFit [54], including datasets such as Food101 [4], SUN397 [53], DF20-Mini [40], Caltech101 [13], CUB-200-2011 [50], ArtBench-10 [30], Oxford Flowers [37], and Stanford Cars [26]. Our base model, the DiT-XL-2-256x256 [39], is pre-trained on ImageNet at $256 \times 256$ resolution, achieving a Fréchet Inception Distance (FID) [16] of 2.27 [2]. The FID is calculated by measuring the distance between the generated images and a test set, serving as a widely used metric for evaluating generative image models' quality. We adhere to the default generation protocol as specified in [54], generating 10K instances with 50 sampling steps (FID-10K). $\beta_{\text{cfg}}$ weight is set to 1.5 for evaluation. For the implemented DiffFit baseline, we follow the optimal settings in [54], which involve enlarging the learning rate $\times 10$ and carefully placing the scale factor to 1 to 14 blocks. For each result, we fine-tune 24K iterations with a batch size of 32 for standard fine-tuning and Diff-Tuning, and a batch size of 64 for DiffFit, on one NVIDIA A100 40G GPU. For each benchmark, we recorded the Relative Promotio of FID between Diff-Tuning and Full Fine-tuning ($\frac{\text{Diff-Tuning} - \text{Full Fine-tuning}}{\text{Full Fine-tuning}}$) to highlight the effectiveness of our method. More implementation details can be found in Appendix B.

---

[2]https://dl.fbaipublicfiles.com/DiT/models/DiT-XL-2-256x256.pt

Table 1: Comparisons on 8 downstream tasks with pre-trained DiT-XL-2-256x256. Methods with "†" are reported from the original Table 1 of [54]. Parameter-efficient methods are denoted by "*".

| Dataset / Method | Food | SUN | DF-20M | Caltech | CUB-Bird | ArtBench | Oxford Flowers | Standard Cars | Average FID |
|---|---|---|---|---|---|---|---|---|---|
| Full Fine-tuning | 10.93 | 14.13 | 15.29 | 23.84 | 5.37 | 19.94 | 16.67 | 6.32 | 14.06 |
| AdaptFormer[†*][7] | 11.93 | 10.68 | 19.01 | 34.17 | 7.00 | 35.04 | 21.36 | 10.45 | 18.70 |
| BitFit[†*][59] | 9.17 | 9.11 | 17.78 | 34.21 | 8.81 | 24.53 | 20.31 | 10.64 | 16.82 |
| VPT-Deep[†*][22] | 18.47 | 14.54 | 32.89 | 42.78 | 17.29 | 40.74 | 25.59 | 22.12 | 26.80 |
| LoRA[†*][21] | 33.75 | 32.53 | 120.25 | 86.05 | 56.03 | 80.99 | 164.13 | 76.24 | 81.25 |
| DiffFit[*][54] | 7.80 | 10.36 | 15.24 | 23.79 | 4.98 | 16.40 | 14.02 | 5.81 | 12.21 |
| **Diff-Tuning** | **6.05** | **9.01** | **13.64** | **23.69** | **3.50** | **13.85** | **12.63** | 5.37 | **11.08** |
| Relative Promotion | 44.6% | 36.2% | 10.8% | 0.6% | 34.8% | 30.5% | 24.2% | 15.0% | 24.6% |

**Results**   Comprehensive results are presented in Table 1 with the best in **bold** and the second underlined. Compared with other baselines, our Diff-Tuning consistently exhibits the lowest FID across all benchmarks, outperforming the standard fine-tuning by a significant margin (relative 24.6% overpass), In contrast, some PEFT techniques do not yield improved results compared to standard fine-tuning. A detailed comparison with DiffFit is discussed in subsequent sections.

## 4.2   Transfer to Controllable Generation

**Setups**   Controlling diffusion models enables personalization, customization, or task-specific image generation. In this section, we evaluate Diff-Tuning on the popular ControlNet [60], a state-of-the-art controlling technique for diffusion models, which can be viewed as fine-tuning the stable diffusion model with conditional adapters at a high level. We test Diff-Tuning under various image-based conditions provided by ControlNet[3], including Sketch [51], Edge [6], Normal Map [49], Depth Map [41], and Segmentation on the COCO [31] and ADE20k [62] datasets at a resolution of $512 \times 512$. We fine-tune ControlNet for 15k iterations for each condition except 5k for Sketch and 20k for Segmentation on ADE20k, using a batch size of 4 on one NVIDIA A100 40G GPU. For more specific training and inference parameters, refer to Appendix B.

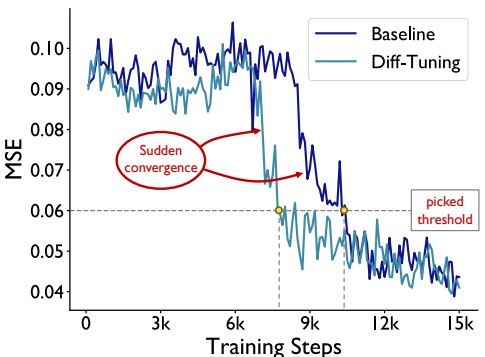

Figure 3: An example of evaluating dissimilarities between conditions (the Normal condition) to infer the occurrence of sudden convergence.

**Evaluation through Sudden Convergence Steps**   Due to the absence of a robust quantitative metric for evaluating fine-tuning approaches with ControlNet, we propose a novel metric based on the sudden convergence steps. In the sudden convergence phenomenon, as reported in [60], ControlNet tends not to learn control conditions gradually but instead abruptly gains the capability to synthesize images according to these conditions after reaching a sudden convergence point. This phenomenon is observable in the showcases presented in Figure 4 throughout the tuning process. We propose measuring the (dis-)similarity between the original controlling conditions and the post-annotated conditions of the corresponding controlled generated samples. As depicted in Figure 3, a distinct "leap" occurs along the training process, providing a clear threshold to determine whether sudden convergence has occurred. We manually select this threshold, combined with human assessment, to identify the occurrence of sudden convergence. The detailed setup of this metric is discussed in Appendix F.

**Results**   As demonstrated in Table 2, Diff-Tuning consistently requires significantly fewer steps to reach sudden convergence across all controlling conditions compared to standard fine-tuning of

---

[3]https://github.com/lllyasviel/ControlNet

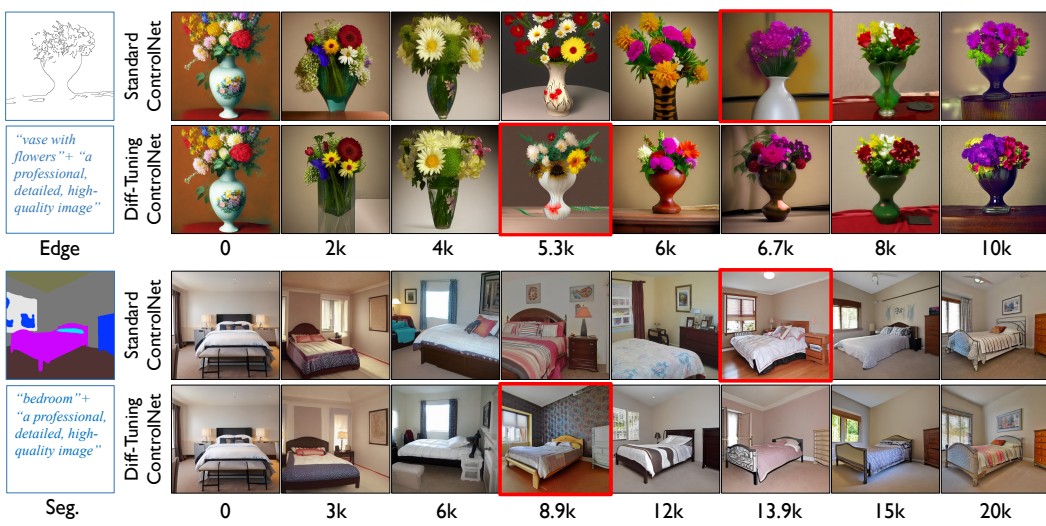

Figure 4: Qualitative compare Diff-Tuning to the standard ControlNet. Red boxes refer to the occurence of "sudden convergence".

Table 2: Sudden convergence steps on controlling Stable Diffusion with 5 conditions.

| Method | Sketch | Normal | Depth | Edge | Seg. (COCO) | Seg. (ADE20k) | Average |
|---|---|---|---|---|---|---|---|
| ControlNet [60] | 3.8k | 10.3k | 9.9k | 6.7k | 9.2k | 13.9k | 9.0k |
| ControlNet +**Diff-Tuning** | **3.2k** | **7.8k** | **8.8k** | **5.3k** | **6.3k** | **8.3k** | **6.6k** |
| Relative Promotion | 15.8% | 24.3% | 11.1% | 20.9% | 31.5% | 40.3% | 24.0% |

ControlNet, indicating a consistent enhancement in the transfer efficiency. In Figure 4, we display showcases from the training process both with and without Diff-Tuning. It is observed that Diff-Tuning achieves sudden convergence significantly faster, enabling the generation of well-controlled samples more quickly. By comparing the images from the final converged model at the same step, it is evident that our proposed Diff-Tuning achieves superior image generation quality.

### 4.3 Discussion on Parameter-Efficient Transfer Learning

The initial motivation behind adapter-based approaches in continual learning is to prevent catastrophic forgetting by maintaining the original model unchanged [20]. These methods conceptually preserve a separate checkpoint for each arriving task, reverting to the appropriate weights as needed during inference. This strategy ensures that knowledge from previously learned tasks is not overwritten. In transfer learning, however, the objective shifts to adapting a pre-trained model for new, downstream tasks. This adaptation often presents unique challenges. Prior studies indicate that PEFT methods struggle to match the performance of full model fine-tuning unless modifications are carefully implemented. Such modifications include significantly increasing learning rates, sometimes by more than tenfold, and strategically placing tunable parameters within suitable blocks [21, 7, 54]. Consider the state-of-the-art method, DiffFit, which updates only the bias terms in networks, merely 0.12% of the parameters in DiT equating to approximately 0.83 million parameters. While this might seem efficient, such a small proportion of tunable parameters is enough to risk overfitting downstream tasks. Increasing the learning rate to compensate for the limited number of trainable parameters can inadvertently distort the underlying pre-trained knowledge, raising the risk of training instability and potentially causing a sudden and complete degradation of the pre-trained knowledge, as observed in studies like [54].

Elastic Weight Consolidation (EWC) [24] is a classic parameter-regularized approach to preserve knowledge in a neural network. We calculate the $L_2$-EWC values, which are defined as EWC $= \|\theta - \theta_0\|^2$, for the tunable parameters in the evaluated approaches [28]. The EWC value quantifies

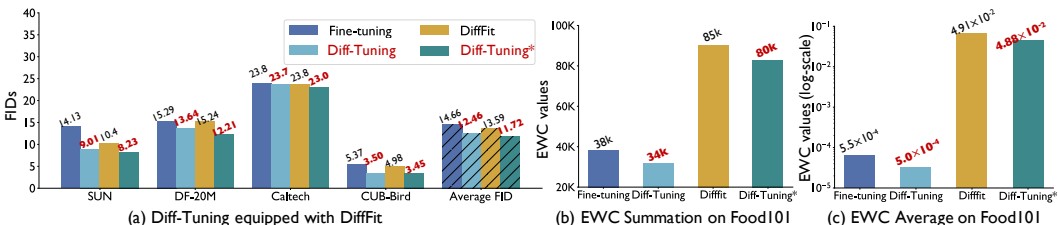

Figure 5: The compatibility of Diff-Tuning with PEFT (a), and catastrophic forgetting analysis (b-c).

how far the fine-tuned model deviates from the pre-trained model, indicating the degree of knowledge forgetting from the perspective of parameter space.

Figure 5(b) reveals that DiffFit leads to EWC values that are 2.42 times larger with only 0.12% tunable parameters, indicating heavy distortion of the pre-trained knowledge. Figure 5(c) illustrates the averaged EWC over tunable parameters, showing that each tunable bias term contributes significantly more to the EWC. In contrast, Diff-Tuning achieves lower EWC values. Diff-Tuning does not explicitly focus on avoiding forgetting in the parameter space but rather harmonizes the chain of forgetting in the parameter-sharing diffusion model and only retains knowledge associated with small $t$ rather than the entire reverse process.

Diff-Tuning can be directly applied to current PEFT approaches, and the comparison results in Figure 5 demonstrate that Diff-Tuning can enhance the transfer capability of DiffFit and significantly improve converged performance.

## 4.4 Analysis and Ablation

**Fine-tuning Convergence Analysis**    To analyze converging speed, we present a concrete study on the convergence of the FID scores for standard fine-tuning, DiffFit, Diff-Tuning, and Diff-Tuning* (DiffFit equipped with Diff-Tuning) every 1,500 iterations in the SUN 397 dataset, as shown in Figure 6(a). Compared to standard fine-tuning and DiffFit, Diff-Tuning effectively leverages the chain of forgetting, achieving a balance between forgetting and retaining. This leads to faster convergence and superior results. Furthermore, the result of Diff-Tuning* indicates that PEFT methods such like DiffFit still struggle with forgetting and overfitting. These methods can benefit from Diff-Tuning.

**Ablation Study**    We explore the efficacy of each module within Diff-Tuning, specifically focusing on knowledge retention and knowledge reconsolidation. We assess Diff-Tuning against its variants where: (1) Only reconsolidation is applied, setting $\xi(t) \equiv 0$ and $\psi(t) = t$; and (2) Only retention is employed, setting $\xi(t) = 1 - t$ and $\psi(t) \equiv 1$. The results, illustrated in Figure 6(b) demonstrate that both knowledge retention and knowledge reconsolidation effectively leverage the chain of forgetting to enhance fine-tuning performance. The FID scores reported on the DF20M dataset clearly show that combining these strategies leads to more efficient learning adaptations.

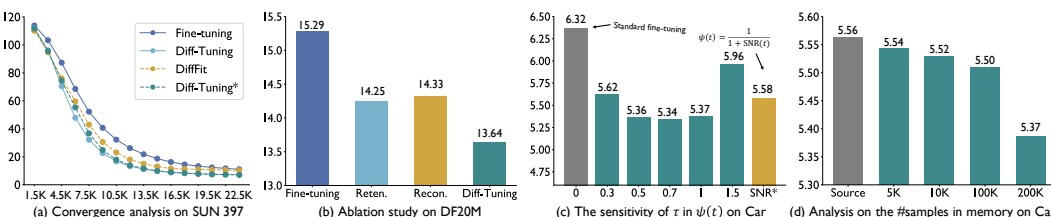

Figure 6: Transfer convergence analysis (a), ablation study (b), and sensitivity analysis (c-d).

**Tradeoff the Forgetting and Retraining with the Chain of Forgetting**    For simplicity and ease of implementation, Diff-Tuning adopts a power function, $\psi(t) = t$, as the default reconsolidation coefficient. To explore sensitivity to hyperparameters, we conduct experiments using various coefficient functions $\psi(t) = t^\tau$ with $\tau$ values from the set $\{0, 0.3, 0.5, 0.7, 1, 1.5\}$, and a signal-to-noise ratio (SNR) based function $\psi(t) = 1/(1 + \text{SNR}(t))$ [9]. Results on the Stanford Car dataset, shown in Figure 6(c), a carefully tuned coefficient can yield slightly better results. To keep the simplicity, we

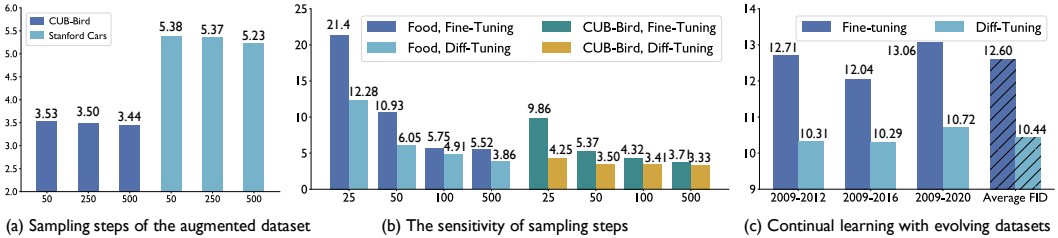

(a) Sampling steps of the augmented dataset (b) The sensitivity of sampling steps (c) Continual learning with evolving datasets

Figure 7: The influence of the quality of the augmented dataset (a), sensitivity with respect to different sampling steps (b), and application of Diff-Tuning in a continual learning setup (c).

keep the default setting $\tau = 1$. Notably, when $\tau = 0$, Diff-Tuning reduces to the standard fine-tuning baseline.

**Analysis on Knowledge Retention** In Diff-Tuning, knowledge is retained using pre-sampled data from pre-trained diffusion models before fine-tuning. We evaluate the impact of varying sample sizes (5K, 50K, 100K, 200K, and the entire source dataset) on the performance of the DiT model on the Stanford Car dataset, as illustrated in Figure 6(d). Notably, using the entire source dataset, which comprises 1.2M ImageNet images, results in suboptimal outcomes. This observation underscores that pre-sampled data serve as a more precise distillation of pre-trained knowledge, aligning with our goal of retraining knowledge rather than merely introducing extra training data.

Beyond the size of the augmented dataset, the quality of the samples can also influence knowledge retention. Therefore, we further analyze the impact of different sampling steps during the pre-sampling stage, as depicted in Figure 7(a). The findings demonstrate that Diff-Tuning exhibits consistent performance, indicating that the knowledge retention process is robust to variations in the sampling process of the augmented dataset.

**Sensitivity to Sampling Steps** We evaluate Diff-Tuning using various sampling parameters, specifically settings of 25, 50, 100, and 500 steps. As depicted in Figure 7(b), there is a consistent improvement in performance across these configurations. Notably, Diff-Tuning shows a more significant enhancement with fewer steps, suggesting that it builds a more precise denoising model compared to standard fine-tuning.

### 4.5 Extend Diff-Tuning to Continual Learning

In dynamic target domains like online systems that continually collect new data, continual learning is essential as models must undergo iterative fine-tuning to adapt to evolving datasets. To evaluate the adaptability of Diff-Tuning in this context, we extend out method and conduct experiments on the Evolving Image Search dataset [63], which includes images from 10 categories collected over three phases: 2009-2012, 2013-2016, and 2017-2020. We apply transfer learning sequentially across these temporal splits, and measure FID on the cumulative test set. For Diff-Tuning, we retain $40\%$ of the initially sampled augmented data and generated the remaining $60\%$ using the fine-tuned model before proceeding to the next phase. As shown in Figure 7(c), the results demonstrate that Diff-Tuning consistently improves and sustains robust performance as the dataset evolves.

## 5 Conclusion

In this paper, we explore the transferability of diffusion models and provide both empirical observations and novel theoretical insights regarding the transfer preferences in their reverse processes, which we term the chain of forgetting. We present Diff-Tuning, a frustratingly simple but general transfer learning approach designed for pre-trained diffusion models, leveraging the identified trend of the chain of forgetting. Diff-Tuning effectively enhances transfer performance by integrating knowledge retention and knowledge reconsolidation techniques. Experimentally, Diff-Tuning shows great generality and performance in advanced diffusion models, including conditional generation and controllable synthesis. Additionally, Diff-Tuning is compatible with existing parameter-efficient fine-tuning methods.

## Acknowledgments and Disclosure of Funding

This work was supported by the National Natural Science Foundation of China (U2342217 and 62021002), the BNRist Project, and the National Engineering Research Center for Big Data Software. We would like to thank our friend Yuchen Zhang for valuable discussions and support for this work.

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

# A  Proofs of Theoretical Explanation in Section 3.1

In this section we provide the formal definitions and proofs for theoretical explanation of chain of forgetting. From the setup of diffusion model training and previous works [23], we suppose the dataset consists of finite bounded samples $D = \{\mathbf{x}_0^{(1)}, \mathbf{x}_0^{(2)}, ..., \mathbf{x}_0^{(n)}\}$, and $f$ is the denoiser to minimize $L(\theta)$ as in Eq. (1) under $\epsilon$-parameterization. For convenience, we first convert the denoiser into $\mathbf{x}_0$-parameterization by $F(\mathbf{x}_t) = \frac{\mathbf{x}_t - \sqrt{1-\alpha_t}f(\mathbf{x}_t)}{\sqrt{\alpha_t}}$ and the objective becomes

$$L = \mathbb{E}_{t,\mathbf{x}_0,\mathbf{x}_t}\left[\|\mathbf{x}_0 - F(\mathbf{x}_t)\|^2\right]. \tag{6}$$

**Ideal Denoiser in Eq. (6).** An ideal denoiser $F$ should minimize the value $F(\mathbf{x}_t)$ for all $t, \mathbf{x}_t$ almost surely, implying an objective for $F(\mathbf{x}_t)$:

$$L_{t,\mathbf{x}_t}(F(\mathbf{x}_t)) = \mathbb{E}_{\mathbf{x}_0 \sim p(\mathbf{x}_0|\mathbf{x}_t)}\left[\|\mathbf{x}_0 - F(\mathbf{x}_t)\|^2\right]. \tag{7}$$

By taking a derivative, it holds that

$$0 = \nabla_{F(\mathbf{x}_t)}L_{t,\mathbf{x}_t}(F(\mathbf{x}_t)) = \mathbb{E}_{\mathbf{x}_0 \sim p(\mathbf{x}_0|\mathbf{x}_t)}\left[-2\left(\mathbf{x}_0 - F(\mathbf{x}_t)\right)\right], \tag{8}$$

and finally,

$$F(\mathbf{x}_t) = \mathbb{E}_{\mathbf{x}_0 \sim p(\mathbf{x}_0|\mathbf{x}_t)}[\mathbf{x}_0] \tag{9}$$

$$= \int_{\mathbf{x}_0} \mathbf{x}_0 \cdot p(\mathbf{x}_0|\mathbf{x}_t)d\mathbf{x}_0 \tag{10}$$

$$= \frac{\int_{\mathbf{x}_0} \mathbf{x}_0 \cdot p_{\mathcal{D}}(\mathbf{x}_0)p(\mathbf{x}_t|\mathbf{x}_0)d\mathbf{x}_0}{p_{\mathcal{D}}(\mathbf{x}_t)} \tag{11}$$

$$= \frac{\int_{\mathbf{x}_0} \mathbf{x}_0 \cdot p_{\mathcal{D}}(\mathbf{x}_0)p(\mathbf{x}_t|\mathbf{x}_0)d\mathbf{x}_0}{\int_{\mathbf{x}_0} p_{\mathcal{D}}(\mathbf{x}_0)p(\mathbf{x}_t|\mathbf{x}_0)d\mathbf{x}_0} \tag{12}$$

$$= \frac{\int_{\mathbf{x}_0} \mathcal{N}\left(\mathbf{x}_t; \sqrt{\alpha_t}\mathbf{x}_0, (1-\alpha_t)\mathbf{I}\right) \cdot \mathbf{x}_0 \cdot p_{\mathcal{D}}(\mathbf{x}_0)d\mathbf{x}_0}{\int_{\mathbf{x}_0} \mathcal{N}\left(\mathbf{x}_t; \sqrt{\alpha_t}\mathbf{x}_0, (1-\alpha_t)\mathbf{I}\right) \cdot p_{\mathcal{D}}(\mathbf{x}_0)d\mathbf{x}_0} \tag{13}$$

$$= \frac{\sum_{\mathbf{x}_0 \in D} \mathcal{N}\left(\mathbf{x}_t; \sqrt{\alpha_t}\mathbf{x}_0, (1-\alpha_t)\mathbf{I}\right) \cdot \mathbf{x}_0}{\sum_{\mathbf{x}_0 \in D} \mathcal{N}\left(\mathbf{x}_t; \sqrt{\alpha_t}\mathbf{x}_0, (1-\alpha_t)\mathbf{I}\right)} \tag{14}$$

*Remark.* This ideal denoiser is exactly the same one as [23] under DDPM-style definition.

**Case when $t \to 0$.** When $t \to 0$, $\alpha_t \to 1$. For simplicity suppose the closest sample to $\mathbf{x}_t$ is unique. Let

$$\mathbf{x}_0^{\text{closest}} = \operatorname{argmin}_{\mathbf{x}_0 \in D}\|\sqrt{\alpha_t}\mathbf{x}_0 - \mathbf{x}_t\|^2, \tag{15}$$

$$d = \min_{\mathbf{x}_0 \in D \setminus \{\mathbf{x}_0^{\text{closest}}\}}\|\sqrt{\alpha_t}\mathbf{x}_0 - \mathbf{x}_t\|^2 - \|\sqrt{\alpha_t}\mathbf{x}_0^{\text{closest}} - \mathbf{x}_t\|^2 > 0, \tag{16}$$

then

$$0 \leq \left\|\frac{\sum_{\mathbf{x}_0 \in D} \mathcal{N}\left(\mathbf{x}_t; \sqrt{\alpha_t}\mathbf{x}_0, (1-\alpha_t)\mathbf{I}\right) \cdot \mathbf{x}_0}{\mathcal{N}\left(\mathbf{x}_t; \sqrt{\alpha_t}\mathbf{x}_0^{\text{closest}}, (1-\alpha_t)\mathbf{I}\right)} - \mathbf{x}_0^{\text{closest}}\right\| \tag{17}$$

$$\leq \sum_{\mathbf{x}_0 \in D \setminus \{x^{\text{closest}}\}} \left\|\frac{1}{\sqrt{2\pi(1-\alpha_t)}}\exp\left(\frac{-\|\sqrt{\alpha_t}\mathbf{x}_0 - \mathbf{x}_t\|^2 + \|\sqrt{\alpha_t}\mathbf{x}_0^{\text{closest}} - \mathbf{x}_t\|^2}{2(1-\alpha_t)}\right)\right\| \tag{18}$$

$$\leq \sum_{\mathbf{x}_0 \in D \setminus \{x^{\text{closest}}\}} \left\|\frac{1}{\sqrt{2\pi(1-\alpha_t)}}\exp\left(-\frac{d}{2(1-\alpha_t)}\right)\right\| \to 0 \tag{19}$$

as $\alpha_t \to 1$, i.e.,

$$\frac{\sum_{\mathbf{x}_0 \in D} \mathcal{N}\left(\mathbf{x}_t; \sqrt{\alpha_t}\mathbf{x}_0, (1-\alpha_t)\mathbf{I}\right) \cdot \mathbf{x}_0}{\mathcal{N}\left(\mathbf{x}_t; \sqrt{\alpha_t}\mathbf{x}_0^{\text{closest}}, (1-\alpha_t)\mathbf{I}\right)} \to \mathbf{x}_0^{\text{closest}}. \tag{20}$$

Similarly

$$\frac{\sum_{\mathbf{x}_0 \in D} \mathcal{N}\big(\mathbf{x}_t; \sqrt{\alpha_t}\mathbf{x}_0, (1-\alpha_t)\mathbf{I}\big)}{\mathcal{N}\big(\mathbf{x}_t; \sqrt{\alpha_t}\mathbf{x}_0^{\text{closest}}, (1-\alpha_t)\mathbf{I}\big)} \to 1, \tag{21}$$

and thus $F(\mathbf{x}_t) \to \mathbf{x}_0^{\text{closest}}$, which completes the proof. Notably, when there are multiple closest samples, through similar analysis it is clear that $F(\mathbf{x}_t)$ converges to their average.

**Case when $t \to T$.** When $t \to T$, $\alpha_t \to 0$, and thus $\mathcal{N}\big(\mathbf{x}_t; \sqrt{\alpha_t}\mathbf{x}_0, (1-\alpha_t)\mathbf{I}\big) \to \mathcal{N}\big(\mathbf{x}_t; \mathbf{0}, \mathbf{I}\big)$, a constant for varying $\mathbf{x}_0$. Bringing this back to Eq. (14) and it holds that

$$F(\mathbf{x}_t) = \frac{1}{n} \sum_{\mathbf{x}_0 \in D} \mathbf{x}_0, \tag{22}$$

which completes the proof.

# B  Implementation Details

We provide the details of our experiment configuration in this section. All experiments are implemented by Pytorch and conducted on NVIDIA A100 40G GPUs.

## B.1  Benchmark Descriptions

This section describes the benchmarks utilized in our experiments.

### B.1.1  Class-conditional Generation Tasks

**Food101 [4]**  The dataset consists of 101 food categories with a total of 101,000 images. For each class, 750 training images preserving some amount of noise and 250 manually reviewed test images are provided. All images were rescaled to have a maximum side length of 512 pixels.

**SUN397 [53]**  The SUN397 benchmark contains 108,753 images of 397 well-sampled categories from the origin Scene UNderstanding (SUN) database. The number of images varies across categories, but there are at least 100 images per category. We evaluate the methods on a random partition of the whole dataset with 76,128 training images, 10,875 validation images and 21,750 test images.

**DF20M [40]**  DF20 is a new fine-grained dataset and benchmark featuring highly accurate class labels based on the taxonomy of observations submitted to the Danish Fungal Atlas. The dataset has a well-defined class hierarchy and a rich observational metadata. It is characterized by a highly imbalanced long-tailed class distribution and a negligible error rate. Importantly, DF20 has no intersection with ImageNet, ensuring unbiased comparison of models fine-tuned from ImageNet checkpoints.

**Caltech101 [50]**  The Caltech 101 dataset comprises photos of objects within 101 distinct categories, with roughly 40 to 800 images allocated to each category. The majority of the categories have around 50 images. Each image is approximately 300×200 pixels in size.

**CUB-200-201 [50]**  CUB-200-2011 (Caltech-UCSD Birds-200-2011) is an expansion of the CUB-200 dataset by approximately doubling the number of images per category and adding new annotations for part locations. The dataset consists of 11,788 images divided into 200 categories.

**Artbench10 [30]**  ArtBench-10 is a class-balanced, standardized dataset comprising 60,000 high-quality images of artwork annotated with clean and precise labels. It offers several advantages over previous artwork datasets including balanced class distribution, high-quality images, and standardized data collection and pre-processing procedures. It contains 5,000 training images and 1,000 testing images per style.

**Oxford Flowers [37]**  The Oxford 102 Flowers Dataset contains high quality images of 102 commonly occurring flower categories in the United Kingdom. The number of images per category range between 40 and 258. This extensive dataset provides an excellent resource for various computer vision applications, especially those focused on flower recognition and classification.

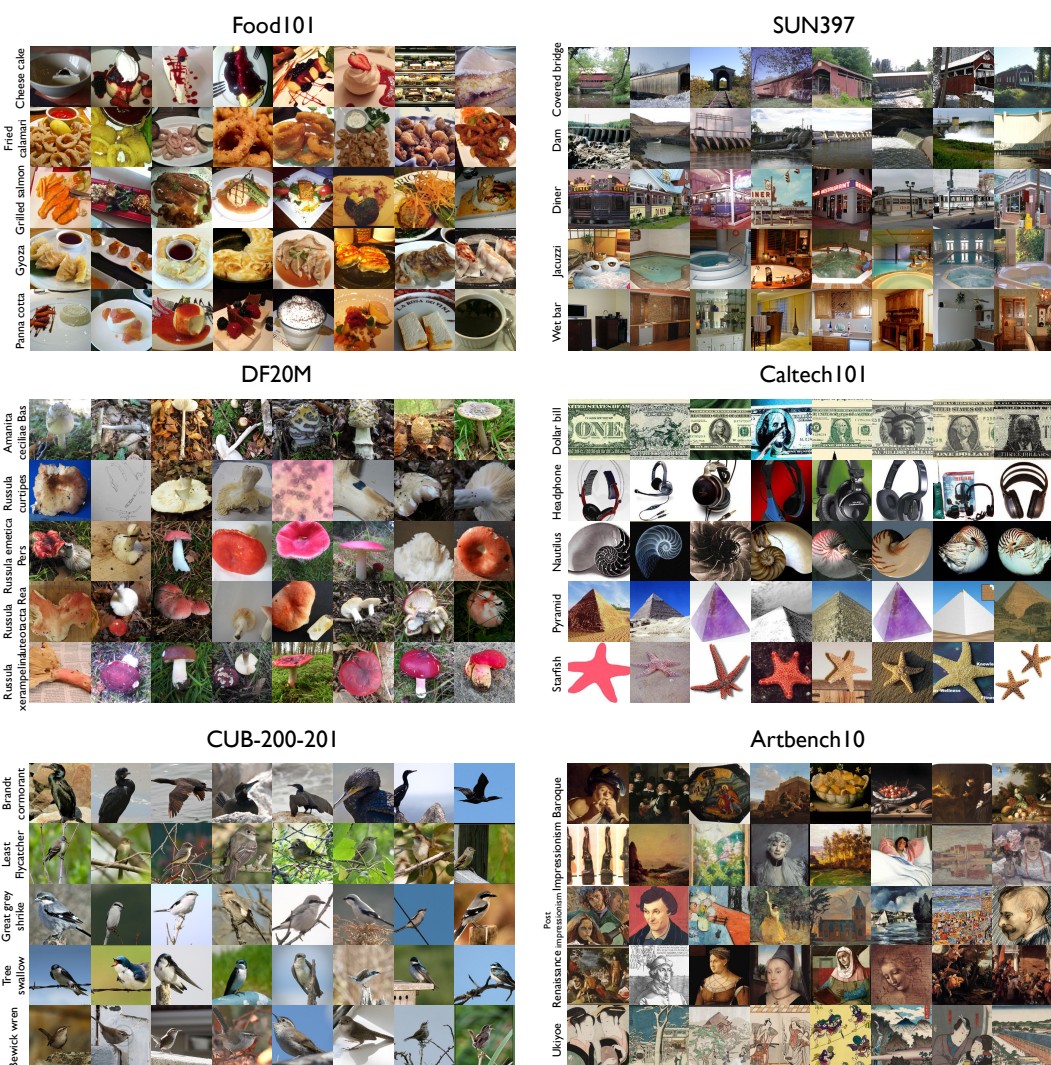

Figure 8: Samples show of different datasets.

**Stanford Cars [26]** In the Stanford Cars dataset, there are 16,185 images that display 196 distinct classes of cars. These images are divided into a training and a testing set: 8,144 images for training and 8,041 images for testing. The distribution of samples among classes is almost balanced. Each class represents a specific make, model, and year combination, e.g., the 2012 Tesla Model S or the 2012 BMW M3 coupe.

### B.1.2 Controllable Generation

**COCO [31]** The MS COCO dataset is a large-scale object detection, segmentation, key-point detection, and captioning dataset. We adopt the version of 2017 where 164k images are split into 118k/5k/41k for training/validation/test. For each image, we randomly select one of its corresponding captions, and use detectors of Canny edge, (binarized) HED sketch, MIDAS depth/normal and Uniformer segmentation implemented in ControlNet to obtain the annotation control, and final construct the dataset of image-text-control pairs for training and evaluation. All controling condition has channel 1 except normal map and segmentation mask with channel 3.

**Ade20k [62]** The Ade20k dataset is a semantic segmentation dataset containing nearly 21k/2k training/validation images annotated with pixel-wise segmentation mask of 149 categories of stuff and objects. For each image, we use the "default prompt" *"a high-quality, detailed, and professional image"* as adopted in ControlNet, and use Uniformer implemented in ControlNet to obtain the segmentation mask as the control to obtain the dataset of image-text-control pairs.

## B.2 Experiment Details

**Detailed Algorithm Process** For all experiments, we first generate images of memory bank with the pre-trained model. We then construct conditioned dataset from memory bank with default condition (unconditional for class-conditional generation task and "default prompt" with generated control for controllable generation task). Considering instability of weighted training loss with $\xi(t)$ and $\psi(t)$, for each iteration, we instead first uniformly sample timestep $t$ according to the property $\xi(t) + \psi(t) = 1$, then determine whether sample $\mathbf{x}_0$ from the downstream dataset or the augmented dataset, and finally calculate the diffusion loss $L(\theta)$ with the simple form. The pseudo-code of the overall algorithm process is shown in Algorithm 1.

---

**Algorithm 1** Pseudo-code of Diff-Tuning

---

**Input:** Downstream dataset $\mathcal{X}$, pre-trained model parameter $\theta_0$
**Output:** Fine-tuned parameter $\theta$.
Collect pre-sampled data $\widehat{\mathcal{X}}^s$ for knowledge reconsolidation using $\theta_0$.
Initialize $\theta \leftarrow \theta_0$.
**while** not converged **do**
   Initialize mini-batch $X = \{\}$.
   **for** $i$ in $\{1, 2, ..., B\}$ **do**
      Sample $t^{(i)}$ from uniform distribution $\mathcal{U}(\{1, 2, ..., T\})$.
      **if** with probability $\xi(t^{(i)})$ **then**
         Sample $\mathbf{x}_0^{(i)} \sim \widehat{\mathcal{X}}^s$.   {Retention loss $L_{\text{retention}}$}
      **else**
         Sample $\mathbf{x}_0^{(i)} \sim \mathcal{X}$.   {Adaptation loss $L_{\text{adaption}}$}
      **end if**
      Add $(\mathbf{x}_0^{(i)}, t^{(i)})$ to mini-batch $X$.
   **end for**
   Calculate

$$L(\theta) \leftarrow \frac{1}{B} \sum_{i=1}^{B} \mathbb{E}_{\boldsymbol{\epsilon}} \left[ \left\| \boldsymbol{\epsilon} - f_\theta \left( \sqrt{\alpha_{t^{(i)}}} \mathbf{x}_0^{(i)} + \sqrt{1 - \alpha_{t^{(i)}}} \boldsymbol{\epsilon}, t^{(i)} \right) \right\|^2 \right]$$

   Update $\theta$ according to $\nabla_\theta L(\theta)$.
**end while**
**return** $\theta$

---

**Hyperparameters** We list all hyperparameters in our experiments in Table 3.

Table 3: Hyperparameters of experiments.

|                        | Class-conditional | Controlled           |
|------------------------|:-----------------:|:--------------------:|
| Backbone               | DiT               | Stable-diffusion v1.5 |
| Image Size             | 256               | 512                  |
| Batch Size             | 32                | 4                    |
| Learning Rate          | 1e-4              | 1e-5                 |
| Optimizer              | Adam              | Adam                 |
| Training Steps         | 24,000            | 15,000               |
| Validation Interval    | 24,000            | 100                  |
| Sampling Steps         | 50                | 50                   |
| Augmented Dataset Size | 200,000           | 30,000               |

## C Knowledge Distillation Implementation of Knowledge Rentention

There exist many possible methods that can achieve a similar effect to Knowledge Retention, provided they adhere to the principle of the chain of forgetting. In this section, we introduce a variant that utilizes knowledge distillation (KD), which avoids the need to pre-sample an augmented dataset and presents a more elegant solution. The KD variant can be formalized as follows:

$$L_{\text{retention}}^{\text{KD}}(\theta) = \mathbb{E}_{t, \boldsymbol{\epsilon}, \mathbf{x} \sim \mathcal{X}} \left[ \xi(t) \left\| f_{\theta_0} \left( \sqrt{\alpha_t} \mathbf{x} + \sqrt{1 - \alpha_t} \boldsymbol{\epsilon}, t \right) - f_\theta \left( \sqrt{\alpha_t} \mathbf{x} + \sqrt{1 - \alpha_t} \boldsymbol{\epsilon}, t \right) \right\|^2 \right], \quad (23)$$

where $f_{\theta_0}$ represents the original pre-trained model. The results, depicted in Table 4, indicate that both techniques should be effective within the framework of Diff-Tuning. The practical decision can be informed by the specific scenario. Below, we summarize reasons why KD was not our initial choice:

- **GPU Memory:** KD requires maintaining a copy of the pre-trained model alongside the fine-tuning model. For large models, this significantly increases memory costs. For example, we can run Diff-Tuning with a batch size of 32 on a single A100 40GB GPU, whereas the KD variant decreases to a batch size of 24.

- **Computational Cost:** KD doubles the forward computation cost by necessitating the matching of output distributions between two models. Notably, pre-computing the KD labels is not feasible due to the inherent noise in diffusion training. For instance, for a batch size of 24, we achieve 2.1 training steps per second, compared to 1.34 for the KD variant.

- **Training Instability:** Transferring a pre-trained model to a domain significantly different from its training data can introduce out-of-distribution corruption during distillation, potentially causing instability in the fine-tuning process. We invested considerable effort to tune a suitable trade-off for the KD loss (0.05), and sometimes the training is easily disrupted due to an unsuitable KD loss setting.

- **Implementation Difficulty:** An elegant KD implementation requires users to be familiar with the code framework and introduces a large design space. In contrast, an augmented replay buffer introduces only a small set of extra source data and changes the training data sampled related to $t$, which is considerably easier to implement across various fine-tuning scenarios.

Table 4: The FID results comparison of the KD varient. KD loss is trade-off by 0.05.

| Methods | CUB-Bird | Standard Cars |
|---|---|---|
| Vanilla Fine-tuning | 5.37 | 6.32 |
| Diff-Tuning with augmented data | 3.50 | 5.37 |
| Diff-Tuning with KD | 3.75 | 4.97 |

## D   Discussion on Time-Reweighting Techniques

Table 5: Comparison with existing timestep weighting strategies.

| Methods | CUB-Bird | Standard Cars |
|---|---|---|
| Vanilla Fine-tuning | 5.37 | 6.32 |
| P2 | 4.68 | 9.31 |
| MIN-SNR-1 | 7.12 | 7.29 |
| MIN-SNR-5 | 9.44 | 9.31 |
| Diff-Tuning | **3.50** | 5.37 |
| Diff-Tuning+P2 | 3.56 | **4.95** |
| Diff-Tuning+MIN-SNR-1 | 3.76 | 5.56 |
| Diff-Tuning+MIN-SNR-5 | 5.84 | 6.50 |

As mentioned above, we ensure $\psi(t) + \xi(t) = 1$ in our Diff-Tuning, which maintains the overall loss in Eq. (5) with the default DDPM weight (uniform weight across timesteps). In our experimental implementation, we uniformly sample timesteps and then sample training data from either the memory buffer or target dataset according to $\psi(t)$ and $\xi(t)$, respectively. Unlike [9, 15], which tailor weights or develop separate models for each timestep. We do not alter the weights of individual timesteps in Diff-Tuning, which is significantly different from existing works.

To further underscore that the weighting strategies from [9] and [15] are orthogonal to our Diff-Tuning approach, we have implemented these strategies alongside our method. The results demonstrate that

while these existing methods can be applied to Diff-Tuning, their integration shows varied impacts on performance.

These results in Table 5 indicate that while existing methods can be adapted to our Diff-Tuning, the performance varies, especially with MIN-SNR strategies, which may not align with the principles of the chain of forgetting, thereby potentially undermining the transfer learning efficacy.

# E   Fine-tuning EDM with Diff-Tuning

In the main expeirments, we have fine-tuned the DiT-XL-2 model (pre-trained using a VP-SDE approach [48]) and Stable Diffusion. We employed the same diffusion training strategy used for the pre-trained models. Additionally, we have evaluated EDM on publicly available repositories[4]. Since EDM incoporates continuous $\sigma$ instead of discrete $t$ in the training state, we extend to use $\psi(\sigma) = \mathrm{cdf}(\sigma)$ and $\xi(\sigma) = 1 - \psi(\sigma)$, aligning our standard Diff-Tuning. Results are shown below:

Table 6: Results of fine-tuning EDM

| Method | EDM: ImageNet 64x64 $\rightarrow$ CIFAR-10 64x64 |
| --- | --- |
| Vanilla | 14.75 |
| Diff-Tuning | 6.04 |

# F   Sudden Convergence of Controlled Generation

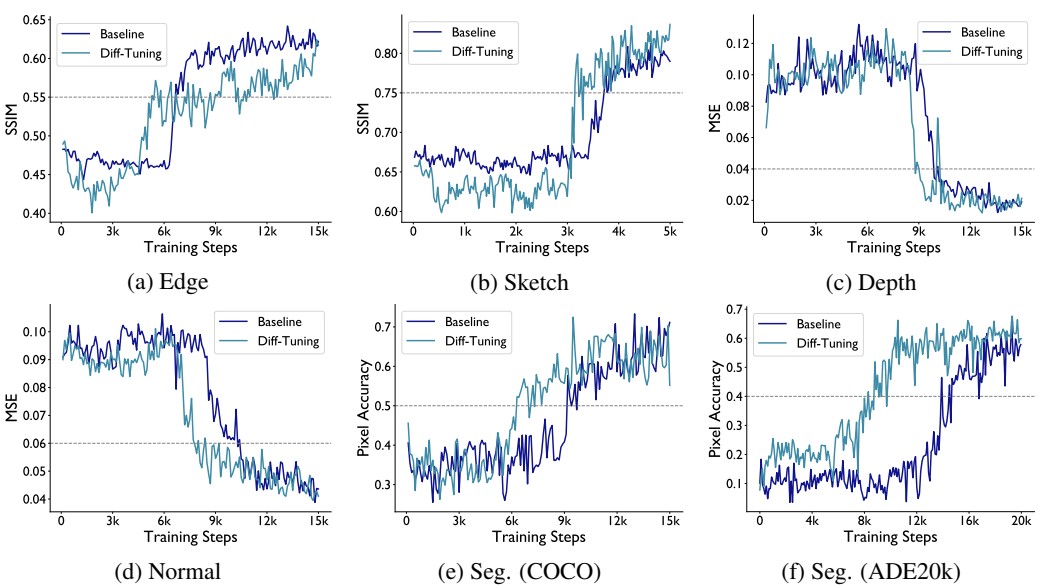

Figure 9: Validation metrics on each task. For Depth and Normal, lower indicates better, and conversely for other tasks.

Sudden convergence is a phenomenon observed when tuning ControlNet [60] due to its specific zero-convolution design. As demonstrated in Figure 4, ControlNet does not gradually learn to adhere to control conditions. Instead, it abruptly begins to follow the input conditions. To identify a simple signal indicating sudden convergence, we pre-collected a validation set of real images and compared the (dis-)similarity between their annotations and the corresponding generated controlled images. Figure 9 illustrates a noticeable "leap" during the training process, providing a clear indicator of sudden convergence. We established thresholds for quantitative evaluation based on test annotation

---

[4]https://github.com/NVlabs/edm

similarity curves and combined them with qualitative human assessment to determine the number of steps as a metric for all tasks.

## F.1 Quantitative Metrics' Details

To efficiently and generally compare the (dis-)similarity between the original controlling conditions and the post-annotated conditions of the corresponding controlled generated samples, we employ the simplest reasonable metrics for each vision annotation, irrespective of task specificities such as label imbalance or semantic similarity. Specifically, we use Structural Similarity Index (SSIM) with Gaussian blurring for sparse classification (Edge and Sketch), mean-square error (MSE) for dense regression (Depth and Normal), and accuracy for dense classification (Segmentation). Detailed settings of the metrics and thresholds are provided in Table 7.

Table 7: Detailed setting of quantitative metrics for controlled generation tasks.

|  | Edge | Sketch | Depth | Normal | Seg. (COCO) | Seg. (ADE20k) |
|---|---|---|---|---|---|---|
| Metric | SSIM w/ Blurring ($\uparrow$) | | MSE ($\downarrow$) | | Pixel Accuracy ($\uparrow$) | |
| Threshold | 0.55 | 0.75 | 0.04 | 0.06 | 0.5 | 0.4 |
| ControlNet | 6.7k | 3.8k | 9.9k | 10.3k | 9.2k | 13.9k |
| ControlNet+Diff-Tuning | 5.3k | 3.2k | 8.8k | 7.8k | 6.3k | 8.3k |

## F.2 More Qualitative Analysis for Human Assessment

We present more case studies to validate the steps for convergence. By generating samples throughout the training process using a consistent random seed, we focus on identifying when and how these samples converge to their corresponding control conditions. As shown in Figure 10, our selected thresholds provide reasonable convergence discrimination across all six tasks, with Diff-Tuning consistently outperforming standard fine-tuning.

**Similarities Alone Are Imperfect**  It is important to note that our proposed similarity metric serves only to indicate the occurrence of convergence and does not accurately reflect sample quality or the degree of control, especially for converged samples. For example, Figure 11 compares generated samples from standard fine-tuning and our Diff-Tuning approach with edge controlling. The generated samples from Diff-Tuning are richer in detail but may score lower on edge similarity metrics, highlighting the limitations of similarity metrics and underscoring the necessity of human assessment.

# G  Limitations and Future Works

Diff-Tuning consistently outperforms standard fine-tuning and parameter-efficient methods, demonstrating its efficacy across various downstream datasets and tasks. This section also discusses some limitations of Diff-Tuning and explores potential future directions to address these limitations.

**Necessity of a Pre-sampled Dataset**  Diff-Tuning involves constructing a pre-sampled augmented dataset for knowledge reconsolidation, which requires additional computational resources. As discussed in Section 4.4, the impact of the sample size indicates that even the smallest set of generated images outperforms baseline methods and the original source data, underscoring the value of this approach. Future work could focus on developing large-scale open-source generated samples and creating more sample-efficient augmented datasets.

**Extra Hyperparameters**  Diff-Tuning introduces additional hyperparameters, $\xi(t)$ and $\psi(t)$, as loss weighting coefficients based on the principle of the chain of forgetting. These introduce extra hyperparameter spaces. As detailed in Section 4.4, our analysis shows that simple designs for these coefficients perform well and robustly. Future research could aim to design more effective weighting coefficients to further enhance performance.

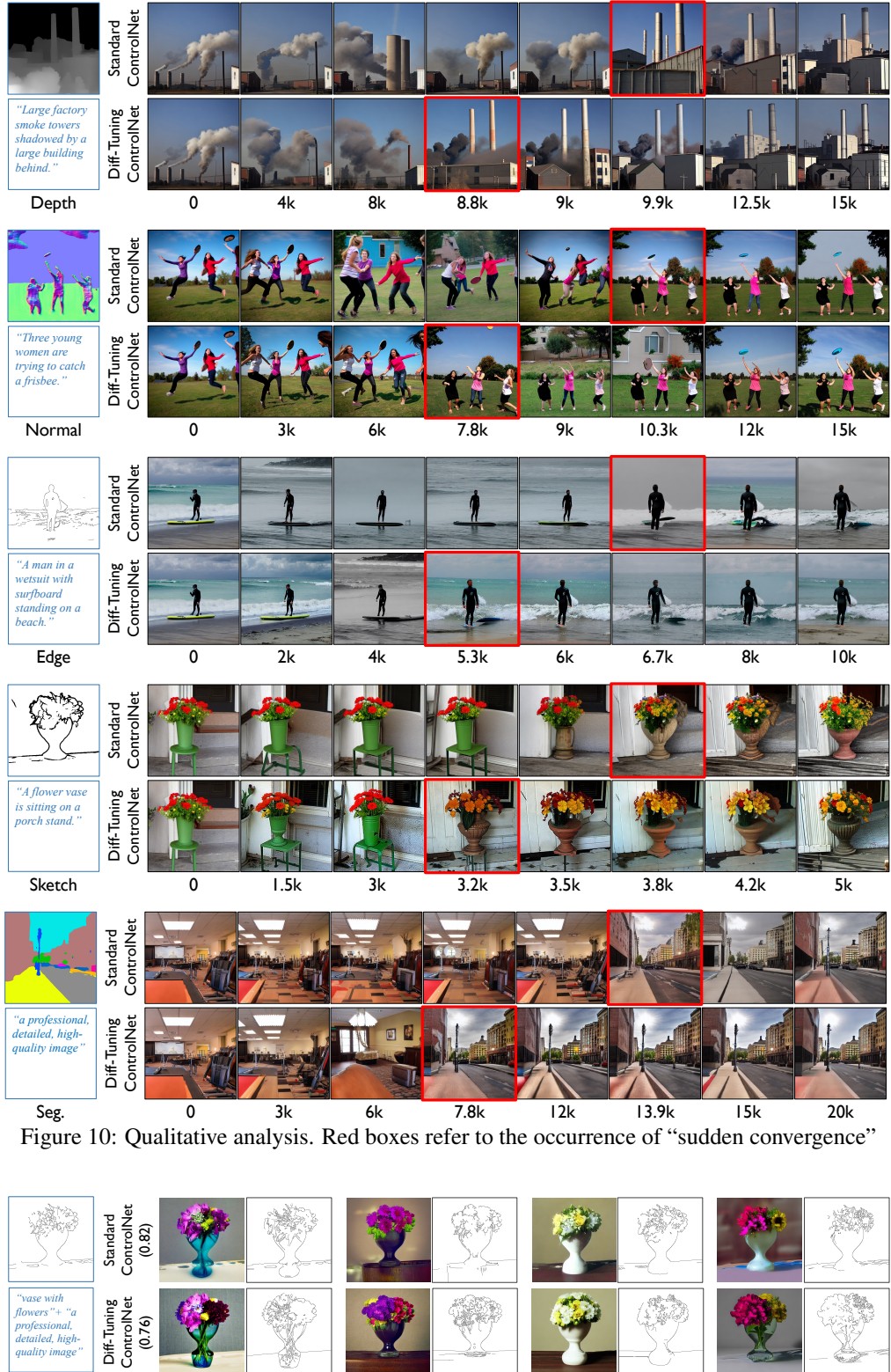

Figure 10: Qualitative analysis. Red boxes refer to the occurrence of "sudden convergence"

Figure 11: Generated images using standard ControlNet *(SSIM of 0.82)* and Diff-Tuning ControlNet *(SSIM of 0.76)*. Analyzing these cases, Diff-Tuning generates images with higher quality and more details hence results in a lower similarity, indicating the limitation of similarity metrics and the necessity of human assessment.

