# OpenReview forum: "Diffusion Tuning: Transferring Diffusion Models via Chain of Forgetting"
_NeurIPS.cc/2024/Conference — NeurIPS 2024 poster_

### Official Review · Reviewer_x2jk · 2024-06-20

**Soundness:** 4
**Presentation:** 3
**Contribution:** 3
**Rating:** 5
**Confidence:** 4

**Summary:**

This paper proposes Diff-Tuning method to encourage the fine-tuned model to retain the pre-trained knowledge. In this method, both pre-trained data and downstream data are used to train the diffusion model. Compared to standard fine-tuning methods, Diff-Tuning enhances the convergence speed and improves the performance. Diff-Tuning can also be used in Conditional Generation. Additionally, Diff-Tuning is compatible with existing parameter-efficient fine-tuning methods.

**Strengths:**

+ The idea of using the pre-trained model to act as a universal denoiser for lightly corrupted data is interesting.
+ Diff-tuning achieved faster training and better performance than standard fine-tuning.
+ This paper provides novel theoretical insights to reveal the principle behind the chain of forgetting. Therefore, this paper is solid.

**Weaknesses:**

+ The novelty of this paper is limited. Many previous works [1,2] find that gradients exist in conflict for diffusion models across timesteps even in the same dataset. But none of them are compared.  **Since the core conclusions of this paper are very similar to them, I think the author needs to highlight the differences.**
1. Efficient Diffusion Training via Min-SNR Weighting Strategy ICCV 2023
2. Addressing Negative Transfer in Diffusion Models. NeurIPS 2023
+ The robustness of Diff-Tuning to sampling algorithms has not been verified, including DDIM [1], DPM-Solver [2], and others. Meanwhile, More diffusion models need to be validated, including VP-SDE [3], Flow Matching [4], EDM [5], etc.
1. Denoising Diffusion Implicit Models ICLR 2021
2. DPM-Solver: A Fast ODE Solver for Diffusion Probabilistic Model Sampling in Around 10 Steps Neurips 2022 O
3. Score-Based Generative Modeling through Stochastic Differential Equations ICLR2021
4.  Flow Matching for Generative Modeling ICLR 2023
5. Elucidating the Design Space of Diffusion-Based Generative Models NeurIPS 2022

+ In the training of the diffusion model, there are different time schedules (Linear, Cosine, etc.), so do I need to keep the same schedule in the Diff-Tuning?

**Questions:**

Please see Weakness. If my question is answered, I will raise my score.

**Limitations:**

Please see Weakness. If my question is answered, I will raise my score.

---

> ### Author Rebuttal · Authors · 2024-08-07
>
> We sincerely thank the reviewer for the efforts in reviewing our paper. Our responses according to the reviewer's comments are summarized as follows.
>
> ---
> > **W1: Concern about the novelty and difference from [1,2].**
>
> Thank you for your insightful thoughts regarding the distinction from existing works. We recognize there may have been some misunderstanding due to the weighted forms in Eq(3) and Eq(4). Our work is distinctively focused on the transferability of pre-trained diffusion models which is significantly different from the approaches taken in [1,2], both in analytical and technical aspects:
> - **Differences in Analysis**: Our work discusses model transferrability from **both theoretical and empirical analysis**, **as detailed in Section 3.1**, which is **never** explored in previous works. [1,2], though also concentrate on varying timesteps $t$, focus on general learning of diffusion models and address the issues of gradient conflicts in multi-task learning.
> - **Differences in Technique Details**: Technically, our proposed Diff-Tuning aims to trade-off **not varying timesteps $t$ but the retention and adaptation**. As stated in Line 194-196, we ensure $\psi(t) + \xi(t) = 1$ in our Diff-Tuning, which **maintains the overall loss in Eq(5) with the default DDPM weight (uniform weight across timesteps)**. And in our experimental implementation, we uniformly sample timesteps and then sample training data from either the memory buffer or target dataset according to $\psi(t)$ and $\xi(t)$, respectively. This contrasts sharply with [1,2], which assign carefully designed weights to different diffusion timesteps $t$ to manage gradient conflicts. From the gradient conflict perspective, we do not alter the weights of individual timesteps; the improvements arise from the transfer preferences we introduce.
>
> Additionally, we present supplementary experiments combining Diff-Tuning with MIN-SNR-$lambda$ ($\lambda = 1,5$) [1].
>
> **Table A': Comparison with MIN-SNR**
> |Methods|cub-200|car|
> |:--:|:--:|:--:|
> |vanilla Fine-tuning|5.32|6.04|
> |MIN-SNR-1 [1]|7.12|7.29|
> |MIN-SNR-5 [1]|9.44|9.31|
> |Diff-Tuning|3.50|5.37|
> |Diff-Tuning+MIN-SNR-1 [1]|3.76|5.56|
> |Diff-Tuning+MIN-SNR-5 [1]|5.84|6.50|
>
> The concerns raised by Reviewer WW5v discuss other existing works, we hope Common Concern #1 in Author rebuttal also helps to address your concerns.
>
> ***From these results, we observe that MIN-SNR [1] does not perform well in transfer learning, since MIN-SNR lacks consideration on transferrability, and its weigting strategy is completely incompatible to the chain of forgetting principle.***
>
> ---
> >**W2: Robustness to sampling algorithms and additional diffusion models**
>
> We agree that evaluating our method with more sampling algorithms and diffusion models would strengthen our findings. To point out, our selected pre-trained models, DiT [5] and Stable Diffusion [6], are the **largest and most representative foundation models** in the diffusion model family, which **ensures the effectiveness** of our methods. Meanwhile, we always follow their default sampling strategies to **ensure the validity** of our experiments. As complement, we further conduct the following experiments to demonstrate the robustness of our method:
>
> - In the original class-conditional experiments, the main results were obtained using 50 uniform DDIM[3] steps. We have extended these experiments to include 100, 250, and 500 DDIM steps, as well as DPM-solver[4]. The results are summarized below:
>
> **Table F: Analysis on the different sampling algorithms**
> | Sampling Algorithm | DDIM-25 | DDIM-50 (default)  | DDIM-100 | DDIM-500 | DPM-solver-20* |
> | :----------------- | :-----: | :------: | :------: | :------: | :-----------: |
> | food-101 (vanilla) |21.4|10.68|5.75|5.52|36.19|
> | food-101 (ours)    |12.28|6.05|4.91|3.86|26.81|
> | cub-200 (vanilla)  |9.86|5.32|4.32|3.71|23.47|
> | cub-200 (ours)     |4.25|3.50|3.41|3.33|14.66|
>
> *Due to the time limit, we do not carefully tune the hyperparameters.
>
> - In the original manuscript, we have fine-tuned the DiT-XL-2 model (pre-trained using a VP-SDE approach [7]) and Stable Diffusion. We employed the same diffusion training strategy used for the pre-trained models. Additionally, we have evaluated EDM on publicly available repositories [9]. Since EDM incoporates continuous $\sigma$ instead of discrete $t$ in the training state, we extend to use $\psi(\sigma)=\text{cdf}(\sigma)$ and $\xi(\sigma)=1-\psi(\sigma)$, aligning our standard Diff-Tuning. Results are shown below:
>
> **Table G: Results of fine-tuning EDM**
> |Method|EDM:ImageNet 64x64$\rightarrow$CIFAR-10 64x64|
> |:---------:|:---------------------------------------------------:|
> |Vanilla|14.75|
> |Diff-Tuning|6.04|
>
> ---
> >**W3: Need to keep the same schedule in the Diff-Tuning?**
>
> Diff-Tuning does not assume any specific scheduling strategy. However, when fine-tuning a model from a pre-trained checkpoint, it is generally preferable to maintain the same schedule as the pre-trained model to avoid potential mismatches. In our experiments, we adhered to this principle by fine-tuning DiT with a linear schedule and Stable Diffusion with its default schedule.
>
>
> ---
> **References**
>
> [1] Efficient Diffusion Training via Min-SNR Weighting Strategy. ICCV 2023. \
> [2] Addressing Negative Transfer in Diffusion Models. NeurIPS 2023.\
> [3] Denoising Diffusion Implicit Models, ICLR 2021\
> [4] DPM-Solver: A Fast ODE Solver for Diffusion Probabilistic Model Sampling in Around 10 Steps, Neurips 2022\
> [5] Scalable diffusion models with transformers, ICCV 2023 \
> [6] Stability. Stable diffusion v1.5 model card, https://huggingface.co/runwayml/stable-diffusion-v1-5, 2022. \
> [7] Score-Based Generative Modeling through Stochastic Differential Equations, ICLR 2021\
> [8] Elucidating the Design Space of Diffusion-Based Generative Models, NeurIPS 2022 \
> [9] EDM repo URL: https://github.com/NVlabs/edm
>
> ---
>
> We hope that our additional clarifications and discussion address your questions and concerns.

---

> > ### Comment · Reviewer_x2jk · 2024-08-08
> >
> > I thank the author for his reply, which resolved my main confusion. Consequently, I have raised my score.

---

### Official Review · Reviewer_y5ux · 2024-07-14

**Soundness:** 4
**Presentation:** 3
**Contribution:** 3
**Rating:** 6
**Confidence:** 4

**Summary:**

This paper explores the fundamental transfer characteristics of diffusion models and observes the monotonous chain of forgetting trend of transferability of diffusion models in the reverse process. It then proposes a simple but effective transfer approach to make the fine-tuned model retain the denoising ability of the pre-trained model close to the generated image, while using domain-specific denoising ability at the beginning of the denoising process. Experimentally, Diff-Tuning can achieve effective improvement on some standard fine-tuning tasks, and also improve the convergence speed of ControlNet.

**Strengths:**

1. Diff-Tuning is simple and easy to follow, and experimentally well realized.
2. the motivation is clear and it explores the transfer capability of the most popular diffusion model, which is novel.
3. this article should make me deepen my understanding of the diffusion model and notice its interesting phenomena in the reverse process.

**Weaknesses:**

1. There are too few metrics on class-conditional generation, while FID is used to extract features with models trained on top of ImageNet, authors use DiT pre-trained on ImageNet to generate samples, which are then added to the training process, it is difficult to say the lower FID actually translates to improved image quality. can you use more metric to show your results?

**Questions:**

1. the authors use a pre-trained diffusion model to preserve denoising ability, would it be useful to use pre-trained domains, i.e. images from ImageNet, instead of augmented images? Although I know that in most cases we can't fully know the pre-trained diffusion model's pre-training data
2. when using a pre-trained diffusion model to generate augmented data, do different sampling methods achieve similar results, and does the choice of sampling steps make a difference?
3. would using a pre-trained diffusion model on ImageNet and then fine-tuning it on small data achieve better results than a diffusion model trained directly on small data? For example, pre-training on ImageNet and then fine-tuning on CIFAR10/100.

**Limitations:**

yes

---

> ### Author Rebuttal · Authors · 2024-08-07
>
> We sincerely appreciate the time and effort you have taken to review our manuscript and for your constructive feedback. We will address the concerns you raised and revise the paper accordingly, as your comments provide valuable insights for improving our work.
>
>
> ---
> > **W1: Use more metric to show the results of class-conditional generation.**
>
> While FID is based on features from models trained on ImageNet, the calculation between the downstream dataset and generated data does not directly reflect any bias from ImageNet training data. In fact, intuitively, using ImageNet might even be perceived as potentially detrimental to the FID value.
>
> Despite FID's widespread use as a metric for evaluating modern generative models, we acknowledge the need for a broader set of evaluations. We agree that additional metrics, such as sFID [1], Precision, and Recall [2], provide a more comprehensive comparison. During the rebuttal phase, we have evaluated Diff-Tuning with these metrics, and the results are as follows:
>
> **Table D: More class-conditional generation metrics evaluated on Stanford Car**
> | Method | FID $\downarrow$ | sFID $\downarrow$ | Precision $\uparrow$ | Recall $\uparrow$  |
> | :-----------| :------: | :-----: | :--: | :--: |
> | Vanilla     | 6.04 | 9.45 | 0.5852 | 0.5873 |
> | Diff-Tuning | 5.37 | 5.43 | 0.6062 | 0.5723 |
>
> These results demonstrate that Diff-Tuning enhances the quality of generation with comparable precision and recall. We also provide showcases under the same random seeds. Please refer to Figure 1 in the attached PDF file.
>
> > **Q1: Would it be useful to use pre-trained domains?**
>
> On the one-hand, using the original pre-training data is indeed applicable to Diff-Tuning. By comparing **Table 1 and Figure 6(d), using the entire pre-training dataset outperforms vanilla fine-tuning, showing the validity of using pre-trained domains.** On the other hand, note that using pre-training dataset may not yield optimal results compared with using generated samples as illustrated in Figure 6(d). We believe this is because that, as discussed in Section 4.4, the fine-tuning stage typically involves fewer training steps and utilizes only a small subset of the pre-training data, whereas the samples generated from a learned model can be more representative than sampling from a large empirical dataset. Hence using pre-trained data, even with accessibility, is sub-optimal.
>
> > **Q2: Do augmented data generated by different sampling methods achieve similar results**
>
> This is an important concern regarding the robustness of Diff-Tuning to different augmented datasets. In our original experiments, the augmented data are generated using the default evaluation protocol provided by the pre-trained model (e.g., the pre-trained DiT-XL-2 was evaluated using DDIM with 250 steps and a CFG of 1.5, so we maintained the same strategy). To further investigate the impact of different sampling methods, we analyze the results with 50 and 500 DDIM steps. These results, in conjunction with Section 4.4 and Figure 6(d), should address your concerns about the impact of different sampling methods on performance.
>
> **Table E: Analysis on the different sampling steps of augmented data**
> | Dataset             | cub-200 | car  |
> | :------------------ | :-----: | :--: |
> | 50 steps            |  3.53   |  5.38|
> | 250 steps (default) |  3.50   |  5.37|
> | 500 steps           |  3.44   |  5.23|
>
>
> > **Q3: Would fine-tuning a pre-trained model achieves better results than training from scratch on small dataset**
>
> The pre-training and fine-tuning paradigm generally outperforms training from scratch, especially for small datasets. Fine-tuning offers numerous advantages, including higher-quality results, lower computational costs, reduced training data requirements, and more efficient training processes.
>
> ---
>
> **References**\
> [1] Generating Images with Sparse Representations, ICML 2021\
> [2] Improved Precision and Recall Metric for Assessing Generative Models, NeurIPS 2019
>
> ---
>
> We hope that our additional clarifications and discussion address your questions and concerns. Please let us know if you have any further concerns.

---

> > ### Comment · Reviewer_y5ux · 2024-08-10
> >
> > Thank you for your reply, authors. This has addressed my main concern. I remain inclined to accept and maintain my score.

---

### Official Review · Reviewer_WW5v · 2024-07-17

**Soundness:** 3
**Presentation:** 3
**Contribution:** 2
**Rating:** 5
**Confidence:** 4

**Summary:**

The paper proposes a new method to fine-tune the pretrained large-scale diffusion models for new tasks. It finds that different time steps of the denoising process of diffusion models have varying transferability. Specifically, the paper finds that "low-noise" time steps close to the end of the denoising process have better transferability. In contrast, the "high-noise" steps responsible for generating the image layout and containing domain knowledge are less transferable. Based on these observations, the paper proposes a novel fine-tuning objective demonstrating better convergence speed than baselines.

**Strengths:**

- The idea is simple without numerous hyperparameters or tricks to work.

- The proposed idea is orthogonal to the main focus of the recent papers that aim to do parameter-efficient fine-tuning, and it can be combined with them as the paper demonstrates in the experiments.

**Weaknesses:**

1. I could not follow the motivation behind the scheme that the paper claims to perform Knowledge Retention with. The main argument of the paper is that "the time steps of the denoising process close to the end of the sampling chain of diffusion models are more transferable." Then, why do we need to sample an augmented dataset to do the Knowledge Retention? A straightforward way can be using knowledge distillation from the original model to the fine-tuned one in the low noise time-steps.

2. Following the previous point, distillation may be better than an augmented dataset considering that the fine-tuning data may have characteristics that are different from the original domain. For instance, (this is just a hypothetical example) the fine-tuning dataset may be on flowers that are mostly red, but the pretaining dataset (ImageNet in the experiments) may not have red flowers. Therefore, for the fine-tuning process, we care more about red flowers, not other types. However, using an augmented dataset can bias the fine-tuning process and lower the model's quality on red flowers. In contrast, distillation on the low-noise time steps may readily preserve the prior knowledge of the model without biasing the fine-tuning process.

3. I appreciate the analysis in Fig. 1, and the formal experimentation of it is a novelty. Yet, some similar phenomena have been shown in the literature before that the paper does not discuss them. [1] shows that a diffusion model usually generates a high-level layout of the image until SNR gets between [0.01, 1], and then, starts to fill in the details. Similarly, eDiff-I shows that changing the text prompt at the low-noise stages of the denoising process process does not change the output image. These experiments in Fig. 1 also show that if we use the original model in the low-noise time steps, the performance does not decrease, which is similar to these papers. I think the paper should include these references in the related work section.

[1] Perception Prioritized Training of Diffusion Models, CVPR 2022.

[2] eDiff-I: Text-to-Image Diffusion Models with an Ensemble of Expert Denoisers, 2023.

**Questions:**

I have the following questions from the paper:

1. When fine-tuning the ImageNet pretrained model on the new task, do you train the class embeddings (used for cross-attention) for the new task as well?

2. Regarding points 1 and 2 in the Weaknesses section, as far as I know, training on images generated by the generative models may degrade the model's performance [1]. Yet, using the whole ImageNet dataset performs worse than using 5k images that are sampled from the pretrained model. Is there any intuition about why this happens?

[1] Self-consuming generative models go mad, ICLR 2024.

**Limitations:**

The authors have pointed to using an augmented dataset as a potential limitation, but as I mentioned in the weakness section, the motivation and intuition of why it should work is not clear.

---

> ### Author Rebuttal · Authors · 2024-08-07
>
> We sincerely appreciate the careful review and insightful suggestions provided by the reviewer. Our responses to the concerns raised are detailed below:
>
> ---
> > **W1&2: The motivation of knowledge retention and a potential varient of Diff-Tuning with knowledge distillation**
>
> Thank you for your insightful comments on the motivation behind knowledge retention and the concerns regarding knowledge distillation (KD). Firstly, it is crucial to emphasize that a novel and critical contribution of our work is the unveiling of the chain of forgetting tendencies, both experimentally and theoretically,  **the theoretical analysis presented in Theorem 1 (Lines 151-162) is never discussed before.** The chain of forgetting tendencies provides a guideline for designing transfer methods.
>
> Utilizing an augmented data buffer in a generative model has been a rational practice since [1], and **KD is a viable implementation only if it adheres to the principle of chain of forgetting**. Both techniques should be effective within the framework of Diff-Tuning. Below, we summarize reasons why KD is not our initial choice:
>
> - **GPU Memory**: KD requires maintaining a copy of the pre-trained model alongside the fine-tuning model. For large models, this significantly increases memory costs. For example, we can run Diff-Tuning with a batch size of 32 on a single A100-40GB GPU, whereas the KD variant decreases to a batch size of 24.
> - **Computational Cost**: KD doubles the forward computation cost by necessitating the matching of output distributions between two models. Notably, pre-computing the KD labels is not feasible due to the inherent noise in diffusion training. For instance, for a batch size of 24, we achieve 2.1 training steps per second, compared to 1.34 for the KD variant.
> - **Training Instability**: Transferring a pre-trained model to a domain significantly different from its training data can introduce out-of-distribution corruption during distillation, potentially causing instability in the fine-tuning process. We invested considerable effort to tune a suitable trade-off for the KD loss (0.05), and sometimes the training is easily disrupted due to an unsuitable KD loss setting.
> - **Implementation difficulty**: An elegant KD implementation requires users to be familiar with the code framework and introduces a large design space. In contrast, an augmented replay buffer introduces only a small set of extra source data and changes the training data sampled related to $t$, which is considerably easier to implement across various fine-tuning scenarios.
>
> Considering scenarios where KD might be preferred over an augmented data buffer, we have also implemented a new variant of knowledge retention incorporating KD. Due to time constraints, we have tuned a preliminary implementation and present the KD results below for a quick overview during the rebuttal phase, and we will revise the methodology section to include this discussion accordingly.
>
> In light of potential settings to prefer KD over an augmented data buffer, we also implement a new varient of knowledge retention incorporating KD. Due to time constraints, we have tuned a preliminary implementation and present the KD results below for a quick overview during the rebuttal phase, and we will revise the methodology section to include this discussion accordingly.
>
> **Table C: The FID results comparison of the  KD varient. (KD loss is trade-off by 0.05)**
> |             Methods             | cub-200 | car  |
> | :-----------------------------: | :-----: | :--: |
> |       Vanilla Fine-tuning       |  5.32   | 6.05 |
> | Diff-Tuning with augmented data |  3.50   | 5.37 |
> |       Diff-Tuning with KD       |  3.75   | 4.97 |
>
>
> > **W3: Similar phenomena noted in previous literature [2,3].**
>
> See Common Concern #1 in Author Rebuttal.
>
>
> > **Q1: Does Diff-Tuning train the class embeddings?**
>
> Yes, we fine-tune the class embeddings in Diff-Tuning using the same methods as in the pre-training stage. To further note, we observed that either reinitializing the class embeddings or directly updating existing ones shows similar performance.
>
> > **Q2: Why using the whole ImageNet dataset performs worse than 5k Images and the intuition behind.**
>
> The slight difference in performance between using the entire ImageNet dataset and 5k augmented images (5.54 vs. 5.52, with vanilla fine-tuning at 6.04) is not statistically significant. As discussed in Section 4.4, one potential explanation is the size of the dataset relative to the fine-tuning duration. Given that fine-tuning only requires 24k steps, a large source dataset is underutilized, where only a fraction of images are iterated (about 1/3 of an epoch). In this context, data sampled from a learned model acts as a form of knowledge distillation, representing the original distribution more effectively than direct sampling from a vast dataset.
>
> [4] explores how synthetic training loops affect generative model performance when the real dataset is fixed. However, this does not directly apply to fine-tuning diffusion models to new domains with new data.
>
> We sincerely thank you for your insightful comments. We will expand the discussion in Section 4.4 in the revised paper accordingly.
>
> ---
> **References**\
> [1] Transferring GANs: generating images from limited data ECCV 2018.\
> [2] Perception Prioritized Training of Diffusion Models, CVPR 2022.\
> [3] eDiff-I: Text-to-Image Diffusion Models with an Ensemble of Expert Denoisers, 2023. \
> [4] Self-consuming generative models go mad, ICLR 2024.
>
> ----
>
> We hope that our responses adequately address your queries and clarify our methodologies. We welcome any further questions or feedback.

---

> > ### Comment · Reviewer_WW5v · 2024-08-13
> > **Official Comment by Reviewer WW5v**
> >
> > I appreciate the authors' rebuttal. The rebuttal addressed most of my concerns, and I raised my score to 5. I did not use a higher score as I think the design choice of using an augmented dataset is not well-motivated. The rebuttal did not provide a compelling answer to my second comment in the weakness section of my initial review. Still, I raised my score as the framework is flexible such that one can replace the augmented dataset with distillation, as the provided experimental results in the rebuttal suggest.

---

### Official Review · Reviewer_a4Zm · 2024-07-21

**Soundness:** 3
**Presentation:** 3
**Contribution:** 2
**Rating:** 6
**Confidence:** 3

**Summary:**

This paper focuses on transfer learning methods for diffusion models. It experimentally demonstrates and provides theoretical insights into how the forgetting trend varies with the diffusion timestep. Based on this observation, they proposes Diff-Tuning. The proposed method introduces objectives for knowledge retention and reconsolidation, ensuring that they are affected decreasingly and increasingly by the diffusion timestep, respectively. The superiority of the proposed method is demonstrated experimentally.

**Strengths:**

The paper is well-written, and the method is well-motivated both experimentally and theoretically.

**Weaknesses:**

* More discussion on how to set the hyperparameters $\xi(t)$ and $\psi(t)$ would be helpful. For example, these parameters perform an weighted sum of the retention and reconsolidation objectives, and combining this with a maximum likelihood analysis could potentially help find hyperparameter selection.

* I am curious if applying the datasets used for pre-training to retention would be better (or worse) than using a pre-sampled dataset and the reason.

* It would be beneficial to discuss how the method can be extended in scenarios where transfer learning is applied multiple times or where the data distribution changes online.

**Questions:**

Please see the Weaknesses part.

**Limitations:**

They provided in the Supplementary section D.

---

> ### Author Rebuttal · Authors · 2024-08-07
>
> We sincerely thank the reviewer for the careful review and insightful suggestions. Our responses to the concerns raised are outlined below:
>
> ---
> > **W1: More discussion on how to set the hyperparameter $\xi(t)$ and $\psi(t)$.**
>
> Thank you for highlighting the importance of hyperparameter selection for $\xi(t)$ and $\psi(t)$. To clarify, **as detailed in Lines 194-196, we set $\xi(t) + \psi(t) = 1$ to ensure uniform weighting across timesteps.** This design simplifies implementation and emphasizes the trade-off between retention and adaptation, without the complexity of a detailed scheduling mechanism. Our approach diverges from conventional MLE-based weighting schedules but can be adapted to include them by modifying $\xi(t)$ and $\psi(t)$ to align with a predefined weight function $w(t)= \frac{\beta_t}{(1-\beta_t)(1-\alpha_t)}$ as discussed in DDPM [1].
>
> - **Robustness of $\xi(t)$ and $\psi(t)$:** As discussed in** Section 4.4 Lines 314-321 and shown in Figure 6(c)**, our results indicate that Diff-Tuning is robust to variations in the design of $\xi(t)$ and $\psi(t)$. This robustness highlights the primary importance of managing the chain of forgetting tendency over specific scheduling details. Our choice, linear to $t$, while simple, proves effective in achieving the desired behavior.
> - **Difference with (MLE-based) weighting schedules:** We recognize that we missed an emphasis in the method details and cause confusion. We here clarify that, as stated **in Lines 194-196, we set $\xi(t) + \psi(t) = 1$ to maintain a consistent weight across all timesteps**, thereby the weighting strategy of $\xi(t)$ and $\psi(t)$ focuses on the trade-off between retention and adaptation, instead of the reweighting across all timestep $t$.
> - **Applicability for any (MLE-based) weighting schedules:** We agree that a maximum likelihood analysis or other sophisticated weighting schedules could potentially aid in the optimal selection of hyperparameters, and present how our Diff-Tuning strategy can be potentially combined with (MLE-based) weighting schedules: Suppose a weighting schedule is defined by the weight function $w(t)$ on the timestep $t$ (such as the MLE-based weighting schedule $w(t)= \frac{\beta_t}{(1-\beta_t)(1-\alpha_t)}$, referring to the original derivation in DDPM [1]), then one can set
> $$
> \xi’(t)=w(t)\xi(t),\quad \psi’(t)=w(t)\psi(t),
> $$
> and replace $\xi(t),\psi(t)$ in Eq.(3)(4) with $\xi’(t),\psi’(t)$ to satisfy $\xi’(t)+\psi’(t)=w(t)$, i.e., in usage of the corresponding weighting schedule. For simplicity, we have retained the most widely used weighting of $w(t)=1$ and do not alter the weight or the distribution of $t$ for fair comparison.
>
> > **W2: Whether using the datasets applied for pre-training during retention would be better (or worse).**
>
> The primary reason we do not use the pre-training data is the typical unavailability in most fine-tuning scenarios. However, using the original pre-training data is indeed feasible with Diff-Tuning. **As discussed in Section 4.4 Lines 325-328 and shown in Figure 6(d), using the entire pre-training dataset may not yield optimal results.** We believe this is because the fine-tuning stage usually involves fewer training steps and utilizes only a small subset of the pre-training data (about 1/3 epoch here), whereas the samples generated from a learned model can be more representative than sampling from a large empirical dataset.
>
> >  **W3: how the method can be extended to scenarios where transfer learning is applied multiple times or where the data distribution changes online.**
>
> The scenario you mentioned aligns with the setting known as continual learning, which is a meaningful extension of Diff-Tuning. It is feasible to extend the replay buffer to collect generated data (or a subset of training data) as the data distribution evolves over time. We conducted experiments on the continual learning dataset Evolving Image Search [2], which consists of images with 10 categories collected in recent years, and splitted into 3 folds: 2009-2012, 2013-2016, and 2017-2020. We perform transfer learning sequentially across these splits, and calculate the FID value in the the accumulated test set. For Diff-Tuning, we maintain $60\%$ of the pre-sampled augmented data and collect other $40\%$ from the fine-tuned model before fine-tuning to the next split. The experimental results are shown below:
>
> **Table B: Continual learning with multiple fine-tuning.**
> | Evolving Dataset    | 2009-2012 (FID) | 2009-2016 (FID) | 2009-2020 (FID) |
> | :------------------ | :-------------: | :-------------: | :-------------: |
> | Vanilla Fine-tuning |      12.71      |     12.04       |    13.06        |
> | Diff-Tuning         |      10.31      |     10.29       |     10.72       |
>
> From the results, is observed that Diff-Tuning demonstrated significant improvements, underscoring the importance of addressing the forgetting phenomenon in continual learning scenarios.
>
> ---
>
> **References**
>
> [1] Denoising Diffusion Probabilistic Models, Neurips 2020. \
> [2] Active Gradual Domain Adaptation: Dataset and Approach, IEEE Transactions on Multimedia 2022.
>
> ---
>
> We hope that our additional clarifications and discussion address your questions and concerns. Please let us know if you have any further concerns!

---

> > ### Comment · Reviewer_a4Zm · 2024-08-13
> >
> > I appreciate the author's response. Most of my concerns have been addressed, and I raise the score.

---

### Author Rebuttal · Authors · 2024-08-07

> **Common Concern #1 (raised by WW5v and x2jk): The comparison to existing literatures [1-4]**

Reviewer WW5v concerns about some similar phenomena have been shown in the existing literatures [1,2]. and Revewer x2jk also points out that [3,4] find gradients conflict across timesteps even in the same dataset. We acknowledge the importance of this issue and will expand the discussion in the related works section of our revised manuscript to highlight our unique contributions further.

It appears there has been a misunderstanding, as the studies cited only superficially resemble our findings. Here are the significant distinctions:

- [1] empirically determines that diffusion models construct a high-level perceptual layout within the SNR range [0.01, 1], and designs a loss weighting strategy based on this range. While [1] focuses on enhancing the efficiency of training diffusion models, it overlooks aspects of transfer learning, which is central to our research.
- [2] finds that diffusion models increasingly rely on the text prompt at higher noise levels. [2] then improves text-conditional generation by deploying a suite of denoisers, each tailored to specific noise levels. However, this approach does not incorporate transfer learning considerations.
- [3,4] focus on addressing gradient conflicts across timesteps $t$ in a general learning context, particularly in the multi-task learning perspective. Their findings and methodologies are **orthogonal** to Diff-Tuning.
- **Our work discusses model transferability from both theoretical and empirical analysis (Section 3.1, Lines 151-162), which has never been explored in previous works.**
- **Technically, our proposed Diff-Tuning aims to trade-off the retention and adaptation rather than strategically weighting varying timesteps $t$**. As stated in Line 194-196, we ensure $\psi(t) + \xi(t) = 1$ in our Diff-Tuning, which maintains the overall loss in Eq(5) with the default DDPM weight (uniform weight across timesteps). In our experimental implementation, we uniformly sample timesteps and then sample training data from either the memory buffer or target dataset according to $\psi(t)$ and $\xi(t)$, respectively. Unlike [1-4], which tailor weights or develop separate models for each timestep. We do not alter the weights of individual timesteps in Diff-Tuning, which is significantly different from existing works.

To further underscore that the weighting strategies from [1] and [3] are orthogonal to our Diff-Tuning approach, we have implemented these strategies alongside our method. The results demonstrate that while these existing methods can be applied to Diff-Tuning, their integration shows varied impacts on performance.

**Table A: Comparison with Existing Timestep Weighting Strategies**
|Methods|cub-200|car |
|:----------------------:|:-----:|:--:|
|Vanilla Fine-tuning|5.32|6.04|
|P2 [1]|4.68|9.31|
|MIN-SNR-1[3]|7.12| 7.29 |
|MIN-SNR-5[3]|9.44| 9.31 |
|**Diff-Tuning**|**3.50**|5.37|
|*Diff-Tuning+P2*[1]|3.56|**4.95** |
|*Diff-Tuning+MIN-SNR-1*[3]|3.76| 5.56 |
|*Diff-Tuning+MIN-SNR-5*[3]|5.84| 6.50 |

These results indicate that while existing methods can be adapted to our Diff-Tuning, the performance varies, especially with MIN-SNR strategies, which may not align with the principles of the chain of forgetting, thereby potentially undermining the transfer learning efficacy.

> **Explanation to the Attached *PDF* File**

In the attached pdf file, we organize all the tables and figures of experiment results and showcases during the whole rebuttal period as a supplement for the reviewers. Here we offer a concise explanation of all the concents.

- **[Table A] (from Reviewer WW5v and x2jk)**: We compare Diff-Tuning with loss weighting strategies, P2 [1] and MIN-SNR [3].
- **[Table B] (from Reviewer a4Zm)**: We extend Diff-Tuning to continual learning scenarios with the Evolving Image Search dataset [5]. We split the dataset into 3 folds by the year of images. We sequentially fine-tune on the 3 folds, and calculate the FID with the accumulated dataset. For Diff-Tuning, we maintain a replay buffer for the augmented data.
- **[Table C] (from Reviewer WW5v)**: We incoporate KD to replace the reconsolidation loss within our Diff-Tuning method. The trade-off for the KD loss is set to 0.05 to avoid instability.
- **[Table D] (from Reviewer y5ux)**: We evaluate with more metrics including sFID [6], Precision and Recall [7] for class-conditional generation tasks.
- **[Table E] (from Reviewer y5ux)**: We use different sampling steps to generate the augmented data.
- **[Table F] (from Reviewer x2jk)**: We use different sampling algorithm (DDIM [8] and DPM-Solver [9]) and steps to sample from the fine-tuned models.
- **[Table G] (from Reviewer x2jk)**: We evaluate EDM [10] pre-trained on ImageNet 64x64 by fine-tuning on CIFAR-10 64x64. For Diff-Tuning, we use $\psi(\sigma)=\mathop{\mathrm{cdf}}(\sigma)$ and $\xi(\sigma)=1-\psi(\sigma)$ to fit continuous $\sigma$ instead of discrete $t$.
- **[Figure A]**: We provide showcases on Standford Cars.


**References**

[1] Perception Prioritized Training of Diffusion Models, CVPR 2022.\
[2] eDiff-I: Text-to-Image Diffusion Models with an Ensemble of Expert Denoisers, 2023. \
[3] Efficient Diffusion Training via Min-SNR Weighting Strategy. ICCV 2023. \
[4] Addressing Negative Transfer in Diffusion Models. NeurIPS 2023. \
[5] Active Gradual Domain Adaptation: Dataset and Approach, IEEE Transactions on Multimedia 2022. \
[6] Generating Images with Sparse Representations, ICML 2021. \
[7] Improved Precision and Recall Metric for Assessing Generative Models, NeurIPS 2019. \
[8] Denoising Diffusion Implicit Models, ICLR 2021. \
[9] DPM-Solver: A Fast ODE Solver for Diffusion Probabilistic Model Sampling in Around 10 Steps, Neurips 2022. \
[10] EDM repo URL: https://github.com/NVlabs/edm.

---

### Decision · Program_Chairs · 2024-09-25

**Decision:**

Accept (poster)

**Comment:**

This paper introduces Diffusion Tuning to transfer learning for diffusion models. By leveraging a new concept termed the "chain of forgetting," the paper proposes an approach to encourage pre-trained diffusion models to retain essential knowledge while discarding noise-related information during fine-tuning. Experiments demonstrate a 26% improvement over conventional fine-tuning across multiple tasks and accelerates ControlNet's convergence by 24%.

The reviewers identify several strengths, including the novelty, clarity and presentation, and experimental results. There are also concerns by some reviewers, including theoretical clarifications and comparisons with existing literature on timestep weighting strategies, comparing with knowledge distillation, and lack of experiment verification to validate the robustness of the method. All the concerns have been discussed and addressed by the authors thorough rebuttals, and the reviewers are positive about the rebuttal.

Given the conceptual novelty, the theoretical foundations, and comprehensive experimental validation, I recommend the paper for acceptance.